# NorSA: Accelerate LLM Decoding via Normalized Sparse Activation

## Abstract

Sparse activation accelerates the decoding of large language models by eliminating redundant computations and reducing memory access during matrix multiplications. Current approaches have potential limitations, as they rely on the strong assumption that "values across different dimensions of hidden states are drawn from independent and identically distributed random variables." Our research challenges this assumption by analyzing how causal dependencies exist between tokens and how correlations exist between different dimensions of hidden states. Building on this insight, we introduce Normalized Sparse Activation (NorSA), a method that accounts for inter-dimensional relationships and integrates contextual information through rotation and norm-based thresholding. NorSA achieves superior performance while maintaining computational efficiency. Experiments across LLaMA, Mistral, and Qwen model series show that NorSA consistently outperforms existing methods. For LLaMA3-8B with 50% activation sparsity, NorSA narrows the perplexity gap to only 0.44 points relative to the dense model, while restricting the zero-shot accuracy decline to a mere 1.23%, surpassing La RoSA by 1.63% and TEAL by 3.9%.

## 1 Introduction

Large Language Models (LLMs) have demonstrated exceptional capabilities across diverse tasks OpenAI (2024); Google (2025); Anthropic (2025), yet their deployment in time-sensitive applications remains challenging due to slow inference speeds. Autoregressive decoding of LLMs is primarily memory-bound, with matrix multiplications in the forward pass consuming substantial computational resources. These operations dominate both inference time and memory bandwidth usage, creating a significant bottleneck. Researchers have explored various acceleration techniques, including quantization Lin et al. (2024); Liu et al. (2025c); Tseng et al. (2024), pruning Han et al. (2016); Frantar & Alistarh (2023); Sun et al. (2024); Ma et al. (2023), and speculative decoding Leviathan et al. (2023), each addressing different aspects of the efficiency challenge.

Sparse activation offers a particularly promising solution by selectively eliminating certain dimensions of activation vectors during forward propagation Li et al. (2023); Liu et al. (2023). This approach reduces both computational workload and memory access requirements by avoiding loading unnecessary weights, thereby accelerating matrix multiplications without requiring architectural changes to the model. Recent studies have revealed that many large language models can be activated sparsely without additional training, sacrificing only acceptable performance.

The key challenge in sparse activation is rapid determination of which dimensions to eliminate while preserving output quality. Deja Vu Liu et al. (2023) addresses this using a small neural network to predict important dimensions, though this introduces additional computational overhead. More recent approaches like CATS Lee et al. (2024), TEAL Liu et al. (2025a) and La RoSA Liu et al. (2025b) employ more efficient mechanisms – threshold-based filtering and top-k selection, respectively. However, both methods rely on the strong assumption that activation values across different dimensions are sampled from independent and identically distributed (i.i.d.) random variables, limiting their effectiveness in real-world scenarios.

In this work, we challenge the i.i.d. assumption underlying previous approaches. Our analysis reveals potential correlations between different activation dimensions, as well as contextual dependencies across tokens. These correlations arise naturally in transformer architectures, where token

representations are conditioned on previous tokens through attention mechanisms, and embedding vectors encode semantic relationships that contradict the independence assumption.

We propose Normalized Sparse Activation (NorSA), a novel approach with three key innovations: (1) determining which dimensions to eliminate based on normalized activation values, seamlessly integrating contextual information into the sparsification process; (2) employing threshold comparison as the sparsification function, which is computationally efficient while allowing flexible parameter allocation for each token's hidden state; and (3) utilizing rotation matrices to reduce linear correlations between dimensions, resulting in more accurate sparsification decisions. Our extensive experiments across LLaMA Touvron et al. (2023); Grattafiori et al. (2024), Mistral Jiang et al. (2023), and Qwen Yang et al. (2024) families demonstrate that NorSA consistently outperforms existing methods, particularly at higher sparsity levels. For LLaMA3-8B at 50% activation sparsity, NorSA maintains performance within 0.44 perplexity points of the dense model while delivering a 1.32× speedup in end-to-end decoding.

## 2 RELATED WORK

Sparse activation is a promising technique for accelerating inference of large language models by reducing unnecessary computations. It leverages the characteristic that many hidden states in a model can be zeroed out with minimal impact on performance. Early works, such as those by Li et al. (2023) and Liu et al. (2023), successfully utilized this natural sparsity in ReLU-based architectures to enhance inference speed.

As large language models evolved to incorporate non-ReLU activations like SwiGLU, researchers pursued new strategies to induce sparsity. Efforts by Mirzadeh et al. (2024) and Zhang et al. (2024) included architectural modifications to foster natural sparsity, while Song et al. (2025) introduced regularization techniques during training to improve sparse activation patterns.

Recently, training-free methods have gained popularity for their applicability to existing models. CATS Lee et al. (2024) introduced a magnitude pruning technique applied to nonlinearly activated intermediate hidden states in feedforward networks. TEAL Liu et al. (2025a) expanded on this approach by applying sparsification to input activations across all model components. La RoSA Liu et al. (2025b)advanced this trend with top-k sparsification and rotation of hidden states.

However, prior training-free methods often depend on the assumption of independently and identically distributed (i.i.d.) activations, which is often not true in practice for large language models. Our work aims to move beyond the limitations of the i.i.d. assumption, refining sparse activation techniques to achieve better performance while maintaining the practicality and applicability of training-free approaches.

## 3 BACKGROUND

### 3.1 SPARSE ACTIVATION

In large language models, linear layers constitute a significant portion of the computational workload. For a linear layer with forward pass formulated as $Y = XW^T = \sum_{i=1}^{d_{in}} X_i \cdot (W_{:,i})^T$, where $X \in \mathbb{R}^{d_{in}}$ represents the input activation vector, $W \in \mathbb{R}^{d_{out} \times d_{in}}$ is the weight matrix, and $Y \in \mathbb{R}^{d_{out}}$ is the output vector, sparse activation offers a compelling opportunity for acceleration.

The key insight is that when any element of $i$-th dimension $X_i = 0$, the corresponding weight column $W_{:,i}$ need not be loaded from memory nor participate in computations. This selective processing directly translates to reduced memory bandwidth usage and fewer arithmetic operations.

Formally, the sparsity of an activation vector $X$ is defined as: $\text{Sparsity}(X) = \frac{1}{d_{in}} \sum_{i=1}^{d_{in}} \mathbf{1}(X_i = 0)$

where $\mathbf{1}(\cdot)$ is the indicator function that equals 1 when the condition is true and 0 otherwise, representing the fraction of zero-valued elements in $X$.

To achieve this sparsity, we apply a sparsification function $\text{Sparsify}(X)$ that transforms the dense activation vector into a sparse one by strategically zeroing out selected elements. The output of a linear layer with activation sparsification is then computed as: $Y' = \text{Sparsify}(X)W^T$

## 3.2 MAGNITUDE-BASED AND TOP-K ACTIVATION SPARSIFICATION

Current approaches to activation sparsification primarily fall into two categories. TEAL employs a magnitude-based approach where elements with absolute values below a threshold $\tau$ are zeroed out:

$$\text{Sparsify}_{\text{TEAL}}(X)_i = \begin{cases} X_i & \text{if } |X_i| > \tau \\ 0 & \text{otherwise} \end{cases} \tag{1}$$

This method offers computational efficiency as it only requires element-wise comparisons against the threshold. The sparsity level can be adjusted through $\tau$, but a critical limitation emerges: TEAL applies a single threshold across all dimensions and all hidden states received by a linear layer. This design choice stems from its underlying assumption that values across different dimensions of activations are sampled from i.i.d. random variables.

La RoSA takes a different approach by implementing a top-k selection strategy, which achieves better performance than simple threshold-based methods:

$$\text{Sparsify}_{\text{La RoSA}}(X)_i = \begin{cases} X_i & \text{if } X_i \text{ is among the } k \text{ largest elements of } X \text{ by magnitude} \\ 0 & \text{otherwise} \end{cases} \tag{2}$$

This method considers all dimension values when determining which dimensions to retain, with the threshold adapting dynamically based on the hidden state. However, it requires sorting operations that become computationally expensive for high-dimensional vectors. Additionally, this approach activates a fixed number of parameters for each hidden state, lacking the flexibility to adapt to varying informational content across different tokens. La RoSA also introduces a rotation of hidden states before sparsification to improve performance. The authors hypothesize that this improvement occurs because, according to the central limit theorem, i.i.d. random variables transformed by an orthogonal matrix tend to converge toward i.i.d. Gaussian random variables, creating a more favorable distribution for sparsification.

## 4 METHOD

### 4.1 MOTIVATION

A fundamental assumption in existing activation sparsification approaches like TEAL and La RoSA is that values across different dimensions of activations can be treated as samples from i.i.d. random variables. However, we question whether this assumption holds true in the context of autoregressive decoding in large language models.

Two key insights challenge this i.i.d. assumption:

First, in autoregressive decoding, each token's probability distribution is conditioned on previously generated tokens, implemented through the attention mechanism. This creates causal relationships between tokens, which naturally extends to their hidden state representations. The hidden state of a token at position $t$ is influenced by all tokens at positions $1$ through $t-1$, establishing dependencies that contradict the independence assumption between the tokens.

Second, the hidden states received by the first layer of the transformer model consist of word embeddings Mikolov et al. (2013), which are fixed constants during inference rather than sampled values. These embeddings are specifically designed with directional and magnitude relationships to encode semantic information. Similar words have similar embedding vectors, and the norm of embedding vectors often correlates with word frequency or semantic importance. These structured relationships directly contradict the i.i.d. assumption between dimensions from the very first layer of the model.

Therefore, we need to relax the i.i.d. assumption and design a sparsification function that accounts for the general case where tokens and dimensions may be correlated. Our approach should satisfy the following criteria:

**Contextual Awareness**: The sparsification function should incorporate contextual information encoded in the complete hidden state, considering all dimension values when deciding whether to retain or zero out any specific dimension.

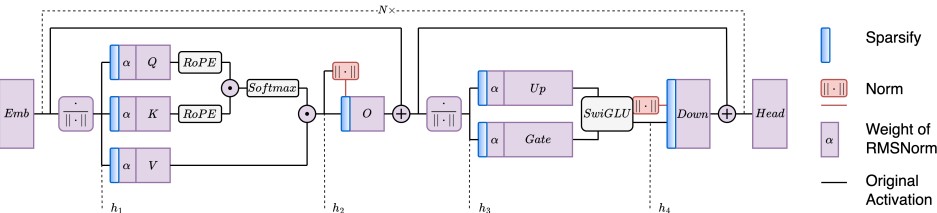

Figure 1: Sparsely activated Transformer architecture with NorSA. Adjacent operations indicate opportunities for kernel fusion or weight merging. RMSNorm weights are folded into subsequent linear layers to reduce norm calculations. The Norm computation in feed forward network receives SwiGLU outputs directly in registers. Detailed kernel fusion is illustrated in Appendix A

**Computational Efficiency**: The computational complexity of the sparsification function should be minimized. Since sparsification and sparse matrix multiplication operate sequentially, the efficiency of the sparsification process directly impacts overall inference speed.

**Flexible Parameter Allocation**: Drawing inspiration from speculative decoding Leviathan et al. (2023) (where a smaller draft model can predict some outputs of the larger model), we recognize that different tokens require different amounts of computational resources. The number of activated parameters for each hidden state during token decoding should be flexibly allocated.

### 4.2 NorSA: Normalized Sparse Attention

To address the limitations of existing approaches and satisfy our design criteria, we propose Normalized Sparse Activation (NorSA), a sparsification method that considers the relative importance of each dimension within the context of the entire activation vector.

The key insight of NorSA is that the significance of a dimension should not be judged by its absolute magnitude alone, but rather by its magnitude relative to the overall scale of the activation vector. This approach naturally incorporates contextual information from all dimensions while maintaining computational efficiency. Formally, we define our sparsification function as:

$$\text{Sparsify}_{\text{NorSA}}(X)_i = \begin{cases} X_i & \text{if } |X_i| > \tau \cdot \|X\| \\ 0 & \text{otherwise} \end{cases} \tag{3}$$

where $\|X\|$ is the norm of vector $X$, and $\tau$ is a threshold ratio hyperparameter that can be set differently for each linear layer. The value of $\tau$ is selected such that:

$$\mathbb{E}_{X \sim p_{\text{data}}}[\text{Sparsity}(\text{Sparsify}_{\text{NorSA}}(X))] = \text{target sparsity} \tag{4}$$

where $X \sim p_{\text{data}}$ indicates that $X$ follows its true data distribution. In practice, $\tau$ is estimated using a calibration dataset that approximates the real distribution of activation vectors during inference.

By adapting the threshold with respect to the norm, NorSA achieves several important benefits:

**Contextual Awareness**: The threshold adapts based on the entire vector's norm, ensuring that dimensions are evaluated within their proper context. In activation vectors with larger overall magnitudes, dimensions need to exceed a proportionally higher absolute threshold to be retained, while in vectors with smaller magnitudes, the threshold is correspondingly lower.

**Computational Efficiency**: Computing the vector norm adds minimal overhead compared to sorting operations required by top-k methods. Furthermore, as shown in Figure 1 the norm calculation can leverage existing pre-norm operations or be fused with the nonlinear activation function in feed forward network, and the element-wise comparison remains computationally efficient.

**Flexible Parameter Allocation**: Different activation vectors naturally retain different numbers of non-zero elements based on their unique distribution of values, allowing for flexible allocation of computational resources.

Figure 2: Sparsely activated transformer architecture incorporating rotation matrices. Rotation matrix $R_1$ is applied to inputs of Q/K/V and Up/Gate projections, while $R_2$ (structured as block diagonal for multi-head attention) is applied to the input of the Output projection. These rotations reduce linear correlations between dimensions of hidden state, enabling more accurate sparsification.

## 4.3 REDUCING LINEAR CORRELATIONS BETWEEN DIMENSIONS VIA ROTATION

While NorSA incorporates information from all dimensions and accounts for inter-token relationships, the correlations between different dimensions within a hidden state still present a challenge. To gain insight into why these inter-dimensional correlations impact sparsification, we examine how removing dimensions affects the norm of output:

$$\|Y\|^2 = \|XW^T\|^2 = XW^TWX^T = \sum_i (W^TW)_{i,i}X_i^2 + \sum_i \sum_{j \neq i} (W^TW)_{i,j}X_iX_j \quad (5)$$

When zeroing out a dimension $X_i$, the impact on the output norm includes both the self-correlation term $(W^TW)_{i,i}X_i^2$ and the inter-dimensional correlation terms $\sum_{j \neq i}(W^TW)_{i,j}X_iX_j$. If these inter-dimensional correlation terms were ignored, the estimation of a dimension's importance by Equation 3 would be inaccurate, potentially missing dimensions with small self-correlation but significant cross-correlations. However, computing these inter-dimensional correlation terms requires $O(d_{in}^2)$ operations, which is prohibitively expensive.

Ideally, we would like to reduce these inter-dimensional correlation terms to make sparsification more accurate and computationally feasible. This can be partly achieved by rotating the activation space to minimize the expectation of linear correlations between dimensions, leveraging the computational invariance of transformer architecture Ashkboos et al. (2024), as shown in Figure 2.

The objective is to find rotation matrices that minimize the sum of square of off-diagonal elements in the expected covariance matrices:

$$\min_R \sum_{i \neq j} \mathbb{E}_{X \sim p_{\text{data}}}[(R^TX^TXR)_{i,j}^2] \quad (6)$$

where $X \sim p_{\text{data}}$ indicates that $X$ follows its true data distribution. In practice, we use a calibration dataset to approximate this distribution.

For rotation matrices that need to be shared across multiple inputs, the optimization objective becomes the sum of the expectations for each input. For example, since $R_1$ is applied to both input of Q/K/V projection ($h_1$) and input of Up/Gate projection ($h_3$), its optimization objective becomes:

$$\min_{R_1} \sum_{i \neq j} \mathbb{E}_{h_1}[(R_1^Th_1^Th_1R_1)_{i,j}^2] + \mathbb{E}_{h_3}[(R_1^Th_3^Th_3R_1)_{i,j}^2] \quad (7)$$

Due to the multi-head attention mechanism, $R_2$ is structured as a block diagonal matrix, where each block corresponds to a query head ($\in R^{d_{head} \times d_{head}}$). For models using Grouped Query Attention (GQA), query heads sharing the same KV heads also share the same block.

For both $R_1$ and $R_2$, we employ the same initialization and refinement strategy. We initialize each rotation matrix using the eigenvector matrix derived from Principal Component Analysis (PCA) on

their respective composite covariance matrices—$h_1^T h_1 + h_3^T h_3$ for $R_1$ and $h_{2_i}^T h_{2_i}$ for each block of $R_2$, where $h_{2_i}$ represents the output of the $i$-th attention head. While these matrices could be further refined using gradient descent with our minimization objectives in Equation 7, our experiments showed that the PCA-derived eigenvector matrices already approach optimal solutions, with additional optimization yielding only marginal improvements. Consequently, we adopt the PCA initialization without further refinement in our final implementation.

## 4.4 HARDWARE AWARE CUSTOMIZED KERNEL AND COMPLEXITY ANALYSIS

To achieve practical acceleration, we implement hardware-optimized kernels specifically designed for GPU execution using Triton OpenAI (2021). We focus on three key components:

**Sparse Matrix-Vector Multiply (GEMV) Kernel**: Building upon insights from Deja Vu Liu et al. (2023) and TEAL Liu et al. (2025a), we develop an efficient sparse GEMV kernel with the following optimizations:(1) Weight matrices are stored in column-major format to facilitate selective column access based on non-zero activation elements. (2) The sparsification function, selective weight loading, and split-K matrix vector multiplication were fused into a single kernel, reducing memory access and kernel launch overhead.

**Fused Norm Computation and Activation**: We fused norm calculation in feed forward network with the SwiGLU activation kernel, further reducing memory accesses overhead.

**Transformer Architecture Optimizations**: We make small modifications to the transformer architecture to enhance efficiency. Specifically, we merge the RMSNorm weights into the subsequent linear layers, ensuring that inputs to Q/K/V and Up/Gate projections have unit norm. This eliminates the need for additional norm calculations during sparsification.

NorSA achieves an excellent balance between effectiveness and efficiency. When excluding the layer-wise rotation matrices $R_1$, its theoretical computational complexity is only marginally higher than TEAL, requiring just one additional norm calculation for the input to the O projection. This stands in stark contrast to La RoSA, which incurs the substantial overhead of sorting operations for every linear layer input. Detailed analysis could be found in Appendix A

The inclusion of the layer-wise rotation matrices $R_1$ introduces an additional computational and space complexity of $O(d_{model}^2)$, which accounts for approximately 7% of the model's overall parameters. The computational overhead is proportionally smaller than the parameter increase, due to the attention mechanism's quadratic complexity.

## 5 EXPERIMENT

**Models and Baselines**: We evaluate NorSA on a set of opensourced large language models, including LLaMA-2 Touvron et al. (2023), LLaMA-3 Grattafiori et al. (2024), Mistral v0.3 Jiang et al. (2023), and Qwen-2.5 Yang et al. (2024) series models. For comparison, we select TEAL and La RoSA as baselines, as they are also training-free approaches, with their performance data quoted from the La RoSA paper. Following La RoSA, we apply sparsification to all token's activations.

**Calibration**: All methods requiring calibration data (including NorSA's threshold and rotation matrices, La ROSA's rotation matrices, and TEAL's thresholds) use the same calibration dataset: 16 randomly sampled sequences of length 2048 from the WikiText-2 Merity et al. (2017) training set.

**Mixed Sparsity Configuration**: We adopt the block-wise greedy optimization technique proposed by TEAL Liu et al. (2025a) to configure sparsity levels for NorSA with a step size of 1% global sparsity. For LLaMA-3-8B, this process takes approximately 26 minutes on an NVIDIA A100.

**Rotation Matrices**: For NorSA, we use the rotation matrices initialized through PCA without further refinement, as our experiments 5.2 showed that the PCA initialization already provides near-optimal performance. For LLaMA-3-8B, it takes approximately 8 minutes on an NVIDIA A100.

**Evaluation**: Model performance was evaluated using both language modeling and downstream tasks. For language modeling, we measure perplexity on the WikiText-2 test set. For downstream tasks (measured using lm-evaluation-harness Gao et al. (2024)), we assess zero-shot performance on multiple benchmarks including ARC-Easy Clark et al. (2018), ARC-Challenge Clark et al. (2018),

Table 1: Zero-shot accuracy on 7 tasks (Acc) and five-shot accuracy on MMLU across different models and sparsity levels.

| | | LLaMA-2 | | | | LLaMA-3 | | | | Qwen-2.5 | | | | Mistral | |
| | | 7B | | 70B | | 8B | | 70B | | 7B | | 72B | | 7B | |
| | Method | Acc 0-shot | MMLU 5-shot | Acc 0-shot | MMLU 5-shot | Acc 0-shot | MMLU 5-shot | Acc 0-shot | MMLU 5-shot | Acc 0-shot | MMLU 5-shot | Acc 0-shot | MMLU 5-shot | Acc 0-shot | MMLU 5-shot |
|---|---|---|---|---|---|---|---|---|---|---|---|---|---|---|---|
| 0% | Dense | 66.69 | 45.85 | 73.66 | 68.80 | 70.05 | 65.26 | 76.29 | 78.71 | 70.34 | 74.21 | 75.58 | 86.08 | 70.44 | 62.34 |
| | TEAL | 65.94 | 44.66 | 73.31 | 67.70 | 69.40 | 63.85 | 73.23 | 74.86 | 69.76 | 73.21 | 75.05 | 85.44 | 70.06 | 61.51 |
| 25% | La RoSA | 66.39 | 45.66 | 73.38 | 68.74 | 69.54 | 64.85 | 76.30 | 78.13 | **70.12** | 73.74 | 75.53 | 85.62 | 70.25 | 61.81 |
| | NorSA | **66.52** | **45.81** | **73.71** | 68.56 | **69.91** | **65.06** | **76.28** | **78.71** | 69.97 | **73.90** | **75.57** | **85.80** | **70.49** | **62.22** |
| | TEAL | 64.92 | 43.46 | 72.47 | 66.78 | 68.14 | 59.84 | 72.24 | 73.23 | 68.61 | 71.44 | 74.65 | 84.80 | 68.76 | 60.17 |
| 40% | La RoSA | 66.15 | 44.66 | 73.31 | 68.16 | 68.79 | 62.61 | 75.41 | 77.62 | 69.67 | 72.33 | 75.35 | 85.53 | 69.44 | 61.15 |
| | NorSA | **66.15** | **45.42** | **73.64** | **68.22** | **69.71** | **64.07** | **75.85** | **78.29** | **69.72** | **73.35** | **75.65** | **85.83** | **70.29** | **61.23** |
| | TEAL | 63.22 | 39.57 | 71.92 | 64.43 | 64.92 | 52.78 | 70.80 | 69.20 | 67.76 | 68.53 | 73.74 | 83.54 | 66.73 | 57.34 |
| 50% | La RoSA | 64.61 | 43.10 | 72.86 | 67.57 | 67.19 | 58.65 | 73.81 | 76.51 | 69.09 | 70.09 | 75.18 | 84.34 | 68.46 | 58.80 |
| | NorSA | **65.49** | **43.25** | **73.31** | **67.73** | **68.82** | **62.12** | **75.34** | **78.15** | **69.26** | **72.28** | **75.45** | **85.66** | **69.84** | **60.41** |

PIQA Bisk et al. (2020), BoolQ Clark et al. (2019), HellaSwag Zellers et al. (2019), OpenBookQA Mihaylov et al. (2018), and WinoGrande Sakaguchi et al. (2020). Additionally, we evaluate five-shot performance on the MMLU benchmark Hendrycks et al. (2021).

## 5.1 MAIN RESULT

**Downstream Task Performance** The results on downstream tasks in Table 1 validate NorSA's effectiveness. NorSA consistently matches or exceeds the performance of baseline methods across all models and sparsity levels on zero-shot tasks. On the more challenging five-shot MMLU benchmark, NorSA demonstrates superior performance retention. Specifically, at 50% sparsity, NorSA maintains strong performance across all evaluated models. For LLaMA-3 8B, NorSA achieves 68.82% accuracy on zero-shot tasks, outperforming both La RoSA (67.19%) and TEAL (64.92%).

Table 2: Perplexity on WikiText-2 test set across different models and sparsity levels.

| | | LLaMA-2 | | LLaMA-3 | | Qwen-2.5 | | Mistral |
| | Method | 7B | 70B | 8B | 70B | 7B | 72B | 7B |
|---|---|---|---|---|---|---|---|---|
| 0% | Dense | 5.47 | 3.32 | 6.13 | 2.85 | 6.85 | 3.87 | 5.31 |
| | TEAL | 6.31 | 3.43 | 6.37 | 3.95 | 6.93 | 3.93 | 5.53 |
| 25% | La RoSA | **5.51** | **3.34** | 6.23 | 2.94 | 6.90 | 3.92 | 5.34 |
| | NorSA | 5.54 | 3.36 | **6.19** | **2.91** | **6.87** | **3.89** | **5.33** |
| | TEAL | 6.40 | 3.61 | 6.83 | 4.45 | 7.20 | 4.07 | 5.98 |
| 40% | La RoSA | 5.64 | 3.44 | 6.60 | 3.37 | 7.10 | 4.06 | 5.44 |
| | NorSA | **5.59** | **3.40** | **6.33** | **3.04** | **6.97** | **3.96** | **5.38** |
| | TEAL | 6.80 | 3.91 | 7.56 | 5.61 | 7.81 | 4.31 | 7.17 |
| 50% | La RoSA | 5.87 | 3.62 | 7.22 | 4.10 | 7.42 | 4.26 | 5.62 |
| | NorSA | **5.72** | **3.47** | **6.57** | **3.33** | **7.13** | **4.07** | **5.46** |
| | TEAL | 7.82 | 4.53 | 9.19 | 9.97 | 9.99 | 7.45 | 8.05 |
| 60% | La RoSA | 6.40 | 3.98 | 8.57 | 5.51 | 8.42 | 4.90 | 6.04 |
| | NorSA | **6.02** | **3.63** | **7.16** | **4.77** | **7.49** | **4.28** | **5.68** |

**Language Modeling Performance** As shown in Table 2, NorSA consistently outperforms both TEAL and La RoSA across most evaluated models and sparsity levels. At 25% sparsity, NorSA achieves perplexity scores that are very close to the dense baseline, indicating only a relatively modest performance gap. The advantages of NorSA become increasingly pronounced at higher sparsity levels. At 60% sparsity, where both baseline methods exhibit obvious degradation, NorSA manages to maintain stable performance. For instance, on the Mistral 7B model, NorSA achieves a perplexity of 5.68 at 60% sparsity, compared to 6.04 for La RoSA and 8.05 for TEAL, effectively demonstrating NorSA's ability to preserve model capabilities even under aggressive sparsification.

These results confirm empirically that NorSA leads to more effective sparsification. By injecting contextual information through norm-based thresholding and reducing inter-dimensional correlations via rotation matrices, NorSA preserves more of the model's original capabilities with minimal computational overhead.

## 5.2 IMPACT OF INTER-DIMENSIONAL LINEAR CORRELATION REDUCTION

To evaluate the effectiveness of our approach to reducing linear correlations between dimensions, we conducted ablation studies on LLaMA-2-7B at 50% activation sparsity, using the same calibration data and evaluation metrics as in our main experiments.

We measure the degree of linear correlation reduction using the ratio of the sum of squared diagonal elements to the sum of all squared elements in the covariance matrix (named self-linear-correlation ratio). Higher values indicate lower inter-dimensional linear correlations, with a maximum value of 100% representing completely linearly uncorrelated dimensions.

Table 3 presents the results for two initialization strategies of rotation matrix: **Identity**: Starting with identity matrices and optimizing using QR reparameterization based optimizer Gu et al. (2025) with constant learning rate of 1e-3. The 0-step configuration is equivalent to the original model without rotation matrices. **PCA**: Starting with PCA-derived matrices and refining them using the same optimizer with constant learning rate of 1e-7.

Table 3: Ablation study on inter-dimensional linear correlation reduction. ppl stands for perplexity.

| Initialize | Step | $h_1$ | $h_2$ | $h_3$ | ppl | 0-shot | MMLU |
|---|---|---|---|---|---|---|---|
| Identity | -100 | 1.1% | 3.5% | 0.8% | 6.657 | 63.46 | 37.98 |
| | 0 | 22.3% | 8.5% | 26.9% | 6.408 | 64.09 | 40.33 |
| | 10 | 44.8% | 9.9% | 52.6% | 6.181 | 64.39 | 40.68 |
| | 50 | 92.8% | 22.3% | 95.7% | 6.056 | 63.98 | 41.42 |
| | 100 | 94.3% | 25.2% | 97.0% | 5.964 | 64.30 | 42.56 |
| PCA | 0 | 97.4% | 27.1% | 98.1% | 5.714 | 64.56 | 43.13 |
| | 100 | 97.5% | 27.1% | 98.1% | 5.707 | 65.70 | 43.26 |

The results reveal several important findings. First, there is a clear relationship between reduced dimensional correlation and improved model performance. Despite only addressing linear correlations, we observe substantial performance gains. As the self-linear-correlation ratio increases from 22.3% to 97.5% for the input to Q/K/V projection ($h_1$) (similarly for $h_2$ and $h_3$), we observe consistent improvements in perplexity (from 6.408 to 5.707) and MMLU accuracy (from 40.33 to 43.26).

Second, PCA initialization provides an excellent starting point that is remarkably close to the optimal solution. The PCA-initialized matrices achieve correlation ratios of 97.4%, 27.1%, and 98.1% for $h_1$, $h_2$, and $h_3$ respectively, resulting in superior performance (5.714 perplexity) compared to even 100 steps of optimization from identity initialization. Further refinement of the PCA-initialized matrices yields only marginal improvements. This finding support our decision to use them without extensive refinement in our final implementation.

Notably, the input to the O projection ($h_2$) maintains high linear correlations between dimensions even after optimization, and its self-linear-correlation ratio remains relatively low (27.1%) due to the constraints imposed by the Grouped Query Attention mechanism. This suggests inherent redundancy across attention heads that cannot be fully eliminated through rotation alone.

### 5.3 EFFECT OF CALIBRATION DATASET

NorSA requires a calibration dataset to determine threshold values, rotation matrices, and mixed sparsity configurations that approximate the true distribution of activations during inference. We investigate the effect of different calibration datasets on LLaMA-2-7B and LLaMA-3-8B using three datasets: WikiText-2, C4 Raffel et al. (2020), and Alpaca Taori et al. (2023) (each consisting of 16 randomly sampled sequences of length 2048).

Results in Table 4 show that while NorSA performance remains relatively stable across different calibration datasets, there are observable differences in effectiveness. Models calibrated with Alpaca data demonstrate consistently advantages over those calibrated with WikiText-2 or C4. Notably, even at 60% sparsity, Alpaca-calibrated models maintain performance closer to their dense baselines. We hypothesize that this performance difference stems from Alpaca's unique characteristics. Since Alpaca consists of instruction data generated by OpenAI's text-davinci-003 engine, it may better align with the output distribution of modern language models during decoding. This finding suggests that further performance gains might be achievable by tailoring calibration datasets to specific deployment scenarios.

Table 4: Comparison across different calibration datasets. ppl stands for perplexity.

| | Dataset | LLaMA-2 7B | | | LLaMA-3 8B | | |
|---|---|---|---|---|---|---|---|
| | | ppl | 0-shot | mmlu | ppl | 0-shot | MMLU |
| 0% | Dense | 5.47 | 66.69 | 45.85 | 6.13 | 70.05 | 65.26 |
| 25% | wikitext2 | 5.54 | **66.55** | **45.82** | 6.19 | 69.87 | 65.06 |
| | c4 | 5.54 | 66.49 | 45.62 | 6.20 | **70.06** | **65.22** |
| | alpaca | **5.53** | 66.43 | 45.61 | **6.17** | 69.93 | 65.05 |
| 40% | wikitext2 | 5.66 | 65.76 | 45.42 | 6.33 | 69.71 | 64.01 |
| | c4 | 5.62 | 65.85 | 44.53 | 6.36 | 69.68 | 64.33 |
| | alpaca | **5.56** | **66.29** | **45.44** | **6.25** | **69.78** | **64.68** |
| 50% | wikitext2 | 5.72 | 64.56 | 43.13 | 6.57 | 68.81 | 62.12 |
| | c4 | 5.78 | 65.62 | 42.88 | 6.61 | 69.20 | 63.09 |
| | alpaca | **5.63** | **66.12** | **44.71** | **6.39** | **69.88** | **63.94** |
| 60% | wikitext2 | 6.02 | 62.04 | 38.21 | 7.16 | 67.32 | 57.96 |
| | c4 | 6.14 | 63.31 | 39.47 | 7.20 | 68.01 | 58.37 |
| | alpaca | **5.78** | **65.00** | **43.37** | **6.73** | **68.88** | **61.78** |

## 5.4 Inference Speedup

**Sparse GEMV Speedup**: Table 5 shows our customized sparse GEMV benchmarks on an NVIDIA A100 using PyTorch 2.4 and Triton 2.3.1. We employed CUDA graphs to reduce kernel launch overhead. The selected matrix dimensions represent common configurations in modern LLM architectures. Results show consistent speedups that scale with sparsity, with larger matrices benefiting more. At 100% sparsity, residual latency exists due to thread block allocation, zero-element multiplications, and atomic additions that could be further reduced through lower-level optimizations.

Table 5: Sparse GEMV latency and speedup compared to torch.matmul on NVIDIA A100.

| Sparsity | Matrix Dimensions ($d_{out}$, $d_{in}$) | | | | | | | | | | | |
|---|---|---|---|---|---|---|---|---|---|---|---|---|
| | (4096, 4096) | | (14336, 4096) | | (4096, 14336) | | (8192, 8192) | | (28672, 8192) | | (8192, 28672) | |
| | $\mu$s | Speedup | $\mu$s | Speedup | $\mu$s | Speedup | $\mu$s | Speedup | $\mu$s | Speedup | $\mu$s | Speedup |
| Dense | 41 | 1.00× | 101 | 1.00× | 92 | 1.00× | 105 | 1.00× | 299 | 1.00× | 292 | 1.00× |
| 0% | 41 | 1.00× | 97 | 1.04× | 92 | 1.00× | 106 | 0.99× | 294 | 1.02× | 291 | 1.00× |
| 25% | 33 | 1.23× | 78 | 1.29× | 74 | 1.23× | 85 | 1.23× | 242 | 1.23× | 235 | 1.24× |
| 50% | 32 | 1.28× | 64 | 1.58× | 63 | 1.45× | 70 | 1.49× | 181 | 1.65× | 188 | 1.55× |
| 75% | 31 | 1.31× | 53 | 1.90× | 53 | 1.73× | 58 | 1.82× | 144 | 2.07× | 147 | 1.98× |
| 100% | 29 | 1.40× | 39 | 2.59× | 40 | 2.28× | 42 | 2.51× | 91 | 3.26× | 97 | 3.01× |

**End to End Decoding Speedup**: Table 6 presents the throughput and the corresponding speedup relative to the dense baseline. All benchmarks were conducted on NVIDIA A100 GPUs (single GPU for 7/8B models and tensor parallelism across two GPUs for 70B models) using the gpt-fast meta (2025) framework for measurement. The end-to-end decoding acceleration aligns with the theoretical speedup expected from matrix multiplication optimization. At 0% sparsity, we observe a performance penalty of 7-10% compared to the dense baseline, primarily due to the overhead introduced by rotation matrices and norm calculations. As sparsity increases, NorSA delivers evident speedups: 1.08× for 7/8B models and 1.17× for 70B models at 25% sparsity, scaling to 1.50× and 1.73× respectively at 75% sparsity. Larger models consistently show greater relative speedup.

Table 6: Throughput and speedup across different models and sparsity levels.

| Sparsity | LLaMA-2-7B | | LLaMA-2-70B | | LLaMA-3-8B | | LLaMA-3-70B | |
|---|---|---|---|---|---|---|---|---|
| | token/s | speedup | token/s | speedup | token/s | speedup | token/s | speedup |
| Dense | 106.45 | 1.00× | 22.18 | 1.00× | 96.37 | 1.00× | 21.66 | 1.00× |
| 0% | 95.89 | 0.90× | 20.75 | 0.94× | 87.45 | 0.91× | 20.14 | 0.93× |
| 25% | 115.06 | 1.08× | 26.03 | 1.17× | 104.14 | 1.08× | 25.37 | 1.17× |
| 50% | 140.93 | 1.32× | 31.66 | 1.43× | 127.12 | 1.32× | 30.80 | 1.42× |
| 75% | 160.02 | 1.50× | 38.35 | 1.73× | 142.09 | 1.47× | 36.72 | 1.69× |

## 6 Conclusion

In this paper, we introduced Normalized Sparse Activation (NorSA), a novel approach to activation sparsification that challenges the independent and identically distributed assumption underlying previous methods. By incorporating contextual information through norm-based thresholding and reducing inter-dimensional linear correlations via rotation matrices, NorSA achieves superior performance while maintaining computational efficiency.

## 7 Limitations and LLM Usage

While NorSA significantly improves upon existing methods, limitations remain. Our contextual incorporation approach is simple due to the stringent performance requirements of sparsification functions, and we address only linear correlations between dimensions due to constraints of current LLM architectures. Future research should explore architectures specifically designed for sparse activation, potentially enabling dynamic token-specific subnetwork activation for greater efficiency.

We acknowledge using LLM to assist with paper writing (grammar correction, expression diversity, table formatting) and code debugging, while emphasizing that all core ideas, experiments, and analyses represent original contributions.

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

## A   HARDWARE AWARE CUSTOMIZED KERNEL

### A.1   FUSED SWIGLU AND SQUARED NORM

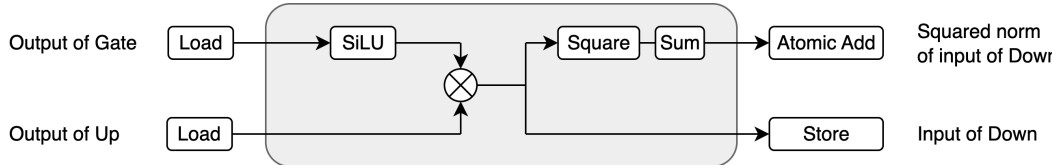

Figure 3: Execution flow diagram of the fused SwiGLU and norm kernel. Gray-shaded areas represent operations performed in registers/cache memory, minimizing data movement and enhancing computational efficiency.

```python
import torch
import triton
import triton.language as tl
from typing import Optional, Tuple, Union

@triton.jit
def silu(x):
    """SiLU activation function: x * sigmoid(x)"""
    return x * tl.sigmoid(x)

norm_configs=[
    # ...
    triton.Config({"BLOCK_SIZE": 256}, num_warps=2),
    # ...
]
@triton.autotune(
    configs=norm_configs,
    key=["hidden_dim"]  # N=input_dim
)
@triton.jit
def swiglu_norm_kernel(
    a_ptr,                     # First input tensor pointer
    b_ptr,                     # Second input tensor pointer
    c_ptr,                     # Output tensor pointer
    norm_squared_ptr,          # Pointer to store squared norm
    hidden_dim,                # Full hidden dimension size
    BLOCK_SIZE: tl.constexpr   # Block size for processing
):
    # Get block index
    block_idx = tl.program_id(0)

    # Calculate start offset for this block
    start_offset = block_idx * BLOCK_SIZE

    # Create offsets and mask for this block
    offsets = start_offset + tl.arange(0, BLOCK_SIZE)
    mask = offsets < hidden_dim

    # Load input values
    a_vals = tl.load(a_ptr + offsets, mask=mask, other=0).to(tl.float32)
    b_vals = tl.load(b_ptr + offsets, mask=mask, other=0)

    # Compute SwiGLU activation: SiLU(a) * b
    silu_a = silu(a_vals)
    c_vals = silu_a.to(b_vals.dtype) * b_vals

    # Store activated values
    tl.store(c_ptr + offsets, c_vals, mask=mask)
```

```python
    # Compute squared values
    c_squared = c_vals.to(tl.float32) * c_vals.to(tl.float32)

    # Sum squared values for this block
    block_sum = tl.sum(c_squared, axis=0)

    # Atomic add to global norm squared
    tl.atomic_add(norm_squared_ptr, block_sum)

def swiglu_norm(a: torch.Tensor, b: torch.Tensor) -> Tuple[torch.Tensor,
    torch.Tensor]:
    """
    fused SwiGLU with squared norm calculation for (1,1,hidden_dim)
    shaped inputs

    Args:
        a: First input tensor of shape (1,1,hidden_dim)
        b: Second input tensor of shape (1,1,hidden_dim)

    Returns:
        Tuple of (activated_tensor, norm_squared)
    """
    # Verify input shapes
    assert a.shape == b.shape, f"Input shapes must match: {a.shape} vs {b
    .shape}"
    assert a.shape[0] == 1 and a.shape[1] == 1, f"Input must have shape
    (1,1,hidden_dim), got {a.shape}"

    # Get hidden dimension
    hidden_dim = a.shape[2]

    # Create output tensor
    c = torch.empty_like(a)

    # Create tensor to store squared norm
    norm_squared = torch.zeros(1, dtype=torch.float32, device=a.device)

    grid = lambda META: (
            triton.cdiv(hidden_dim, META["BLOCK_SIZE"]),
    )

    # Launch kernel with one block per BLOCK_SIZE elements
    swiglu_norm_kernel[grid](
        a,
        b,
        c,
        norm_squared,
        hidden_dim=hidden_dim,
    )
    return c, norm_squared
```

## A.2 SPARSE MATRIX-VECTOR MULTIPLY (GEMV) KERNEL

Our sparse GEMV implementation is optimized for GPU execution using Triton. The kernel design leverages several key insights to maximize performance while implementing NorSA's sparsification.

Figure 4 illustrates the execution flow of our sparse GEMV kernel, highlighting how operations performed in registers or cache memory (gray-shaded areas) minimize global memory accesses. The weights are selectively loaded based on the sparsification mask, significantly reducing memory bandwidth requirements and computational overhead.

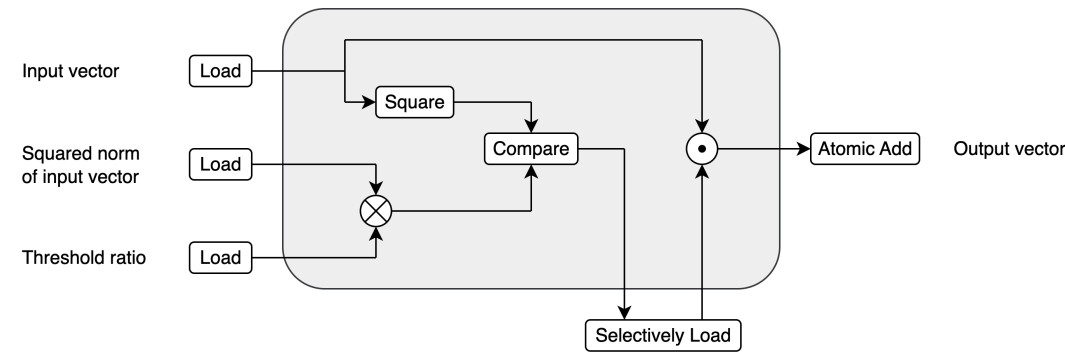

Figure 4: Execution flow diagram of the sparse GEMV kernel. Gray-shaded areas represent operations performed in registers/cache memory. Weights are stored in column-major format and selectively loaded based on the sparsification mask, reducing memory bandwidth requirements and computational overhead. In our implementation, we use squared norm values and compare against squared element values, which is mathematically equivalent to the formulation described in the Method section.

The kernel processes input vector $X$ and weight matrix $W$ to produce output vector $Y$, where $W$ is stored in column-major format to facilitate selective column loading based on the sparsification mask. Each thread block processes a segment of $X$ and computes partial results for a segment of $Y$. The final output for each segment of $Y$ is the sum of contributions from all thread blocks processing the same output segment.

Our implementation computes the squared norm of the input vector and compares each element's squared value against the threshold (proportional to the squared norm), which is mathematically equivalent to the formulation in our method section but computationally more efficient. This approach requires only a single pass through the data with no sorting or additional memory accesses.

Through extensive autotuning, we determined that the optimal configuration uses block dimensions of BLOCK_M = 128 and BLOCK_N = 64 with 4 warps and 2 stages on NVIDIA A100 GPU. The thread blocks can execute completely in parallel, with the latency of parallel block-wise matrix-vector multiplication being $O(\text{BLOCK\_M} \cdot \text{BLOCK\_N})$.

Additionally, NorSA requires computing the squared norm for the input to the O projection. This calculation has a latency of $O(\log \text{Norm\_BLOCK\_N})$ in parallel computing scenarios due to the reduction operation required within thread blocks.

Unlike top-k sparsification that requires sorting operations, our NorSA approach incorporates contextual information and considers all dimension values without the need for sorting. While our block-wise matrix multiplication has a practical latency of $O(\text{BLOCK\_N} \cdot \text{BLOCK\_M})$ (e.g., $O(128 \cdot 64)$), sorting would introduce substantial latency of $O((\log N)^2)$ in parallel scenarios, where $N$ is the full dimension of the input vector (e.g., $O((\log 14336)^2)$). This theoretical complexity disadvantage is further exacerbated in practice because dimensions like the 14,336-dimensional input of the Down projection in LLaMA-2-7B exceed the processing capacity of a single thread block, necessitating inter-block communication/synchronization and significantly increasing latency.

```
1  import torch
2  import triton
3  import triton.language as tl
4  from typing import Optional
5  import os
6  os.environ['TRITON_PRINT_AUTOTUNING'] = '1'
7
8  def init_to_zero(*names):
9      def init_func(nargs):
10         for name in names:
11             nargs[name].zero_()
```

```python
12      return init_func
13
14  norm_configs=[
15      # ...
16      triton.Config({"Norm_BLOCK_N": 256}, num_warps=2, pre_hook=
        init_to_zero("Y")),
17      # ...
18  ]
19  @triton.autotune(
20      configs=norm_configs,
21      key=["N"]   # N=input_dim
22  )
23  @triton.jit
24  def norm_squared_kernel(
25      Y,   # Output pointer (1 x 1)
26      X,   # Input pointer (1 x 1 x N)
27      N,   # N=input_dim
28      Norm_BLOCK_N: tl.constexpr,  # Block size for input_dim (N)
29  ):
30      # Block indices
31      start_n = tl.program_id(0)
32      input_indices = start_n * Norm_BLOCK_N + tl.arange(0, Norm_BLOCK_N)
33      X_ptr = X + input_indices
34      Y_ptr = Y
35
36      # Load input and create sparsity mask
37      input_mask = input_indices < N
38
39      x = tl.load(X_ptr, mask=input_mask, eviction_policy='evict_last')
40      acc = tl.sum(x.to(tl.float32) * x.to(tl.float32))
41      tl.atomic_add(Y_ptr, acc)
42
43
44  configs=[
45      # ...
46      triton.Config({"BLOCK_M": 128, "BLOCK_N": 64}, num_warps=4, pre_hook=
        init_to_zero("Y")),
47      # ...
48  ]
49
50  @triton.autotune(
51      configs=configs,
52      key=["N", "M", "threshold_ratio"]  # N=input_dim, M=out_dim
53  )
54  @triton.jit
55  def sparse_gemv_kernel(
56      Y,   # Output pointer (1 x 1 x M)
57      A,   # Weight pointer (M x N, column-major)
58      X,   # Input pointer (1 x 1 x N)
59      threshold_ratio,  # Threshold for sparsity
60      norm_squared_ptr,     # Squared norm of X
61      N, M,              # N=input_dim, M=out_dim
62      BLOCK_N: tl.constexpr,  # Block size for input_dim (N)
63      BLOCK_M: tl.constexpr,  # Block size for out_dim (M)
64  ):
65      # Block indices
66      start_n = tl.program_id(0)
67      start_m = tl.program_id(1)
68      input_indices = start_n * BLOCK_N + tl.arange(0, BLOCK_N)
69      output_indices = start_m * BLOCK_M + tl.arange(0, BLOCK_M)
70
71      # A is column major: A[output][input] = A + input * M + output
72      A_ptr = A + (input_indices[None, :] * M + output_indices[:, None])
73      X_ptr = X + input_indices
74      Y_ptr = Y + output_indices
```

```
75
76      # Load input and create sparsity mask
77      input_mask = input_indices < N
78      output_mask = output_indices < M
79
80      norm_squared = tl.load(norm_squared_ptr)
81
82      x = tl.load(X_ptr, mask=input_mask, eviction_policy='evict_last')
83      x_squared = x * x
84      sparse_mask = (x_squared > threshold_ratio * norm_squared) &
        input_mask
85
86      # Load weights using sparsity mask
87      a = tl.load(A_ptr, mask=output_mask[:, None] & sparse_mask[None, :],
88                  other=0.0, eviction_policy='evict_first')
89
90      # Compute sparse matrix-vector product
91      acc = tl.sum(a.to(tl.float32) * x.to(tl.float32)[None, :], axis=1)
92
93      # Atomic add to output
94      tl.atomic_add(Y_ptr, acc, mask=output_mask)
95
96
97  def sparse_gemv(
98      x: torch.Tensor,
99      weight: torch.Tensor,
100     threshold_ratio: float,
101     norm_squared: Optional[torch.Tensor] = None,
102 ) -> torch.Tensor:
103     """
104     Compute y = weight @ sparse(x), where weight is stored in column-
        major format.
105
106     Args:
107         x: Input tensor [1, 1, input_dim]
108         weight: Weight matrix [out_dim, input_dim] (column-major)
109         threshold_ratio: Sparsity threshold ratio
110         norm_squared: Optional, squared norm of input vector. If None, it
         will be computed
111
112     Returns:
113         output: Result tensor [1, 1, out_dim]
114     """
115     M, N = weight.shape  # M=out_dim, N=input_dim
116     batch_size, seq_len, input_dim = x.shape
117
118     input_vec = x.contiguous()
119
120     if weight.stride(0) != 1:
121         raise ValueError("Weight matrix must be column-major (first
        dimension contiguous)")
122
123     # Compute squared norm (if not provided)
124     if norm_squared is None:
125         norm_squared = torch.empty(1, device=x.device, dtype=torch.
        float32)
126         grid = lambda META: (
127             triton.cdiv(N, META["Norm_BLOCK_N"]),
128         )
129         norm_squared_kernel[grid](
130             norm_squared,
131             input_vec,
132             N
133         )
134
```

```python
        # Create output tensor
        output_vec = torch.empty(
            1, 1, M,  # Output dimension is the row count of weight matrix
            device=x.device,
            dtype=torch.float32,
        )

        # Define grid size
        grid = lambda META: (
            triton.cdiv(N, META["BLOCK_N"]),  # Number of blocks in input
        dimension
            triton.cdiv(M, META["BLOCK_M"]),  # Number of blocks in output
        dimension
        )

        # Call the optimized kernel
        sparse_gemv_kernel[grid](
            output_vec,          # Output pointer
            weight,              # Weight matrix (column-major)
            input_vec,           # Input vector
            threshold_ratio,     # Sparsity threshold ratio
            norm_squared,        # Squared norm of input vector
            N,                   # Input dimension
            M,                   # Output dimension
        )

        output_vec = output_vec.to(dtype=x.dtype)

        return output_vec
```

## B  FUTURE WORKS

In this section, we outline several promising directions for future research that could build upon the foundations established by NorSA. We have not yet conducted experiments to validate these approaches, but we believe they represent valuable avenues for further exploration.

### B.1  POTENTIAL COMBINATION WITH SPECULATIVE DECODING

Speculative decoding has emerged as a promising approach to accelerate autoregressive generation in large language models. The core idea is to use a smaller, faster model (the "draft" model) to generate candidate tokens in parallel, which are then verified by the larger target model in a single forward pass. This approach increases computational intensity (the ratio of computation to memory access) and effectively amortizes the cost of the large model's forward pass across multiple tokens.

The theoretical maximum speedup of speculative decoding is bounded by the ratio between the generation speeds of the draft model and the target model. If the draft model can generate tokens $k$ times faster than the target model, and all speculative tokens are accepted, the maximum achievable speedup approaches $k$.

Sparse activation presents a compelling opportunity to enhance speculative decoding by increasing the speed of the draft model. By applying NorSA to the draft model, we can accelerate its token generation while maintaining sufficient accuracy for effective speculation. This would directly increase the theoretical maximum speedup achievable through speculative decoding.

### B.2  POTENTIAL COMBINATION WITH MIXTURE OF EXPERTS

Mixture of Experts (MoE) is a powerful approach for scaling language models by conditionally activating only a subset of model parameters for each token. In MoE architectures, the feed-forward network is replaced with multiple "expert" networks, and a routing mechanism (typically a learned gate) determines which experts process each token.

While MoE sparsifies at the granularity of experts—entire blocks of parameters organized as continuous rows and columns—our NorSA approach operates at a finer granularity by sparsifying individual dimensions of activation vectors. Furthermore, MoE typically focuses on sparsifying the intermediate dimensions of MLPs (the outputs of up/gate projections and inputs to down projections), whereas NorSA sparsifies inputs to all linear layers throughout the model.

These complementary approaches could be combined in several ways:

**Dual-level sparsity**: NorSA could be applied within each expert of an MoE architecture, providing an additional level of sparsification beyond expert selection. This would allow for more efficient computation even within the selected experts.

**Informed sparsification**: The gating information from MoE routing mechanisms could be incorporated into NorSA's sparsification function, potentially providing additional context about which dimensions are likely to be important for a given token.

**Explicit sparsity modeling**: Our work with NorSA demonstrates that dense pretrained models can be effectively sparsified post-training without explicit sparsity modeling during pretraining. However, drawing inspiration from MoE's explicit modeling of sparsity, future work could explore pretraining approaches that explicitly encourage activation patterns amenable to NorSA sparsification, potentially enabling even higher

The combination of these approaches could lead to models that benefit from both the architectural sparsity of MoE and the activation sparsity of NorSA, potentially offering multiplicative efficiency gains while maintaining model quality.

## C  DETAILED EXPERIMENT RESULT

Table 7: Performance comparison of different sparsification methods on LLaMA-2-7B.

| Sparsity | Method | WinoGrande | PiQA | OBQA | HellaSwag | BoolQ | ARC-E | ARC-C | Avg | MMLU |
|---|---|---|---|---|---|---|---|---|---|---|
| 0% | Dense | 69.14 | 79.05 | 44.20 | 75.98 | 77.71 | 74.54 | 46.25 | 66.69 | 45.85 |
| 25% | CATS | 67.64 | 77.80 | 41.20 | 75.75 | 70.55 | 69.70 | 43.34 | 63.71 | 42.76 |
| | TEAL | 67.95 | 78.18 | **44.00** | 75.20 | 76.71 | 73.91 | 45.64 | 65.94 | 44.66 |
| | LaRoSA | 68.43 | **79.05** | **44.00** | **75.84** | 77.22 | **74.66** | 45.56 | 66.39 | 45.66 |
| | NorSA | **69.38** | 79.00 | 43.20 | 75.75 | **77.80** | 74.49 | **46.08** | **66.52** | **45.81** |
| 40% | CATS | 56.12 | 66.87 | 32.80 | 52.28 | 63.21 | 44.32 | 31.23 | 49.55 | 24.67 |
| | TEAL | 66.93 | 77.96 | 42.60 | 74.09 | 75.90 | 73.31 | 43.68 | 64.92 | 43.46 |
| | LaRoSA | 68.90 | 78.18 | **44.20** | **75.35** | 76.85 | 73.86 | 45.73 | 66.15 | 44.66 |
| | NorSA | **69.38** | **78.56** | 43.60 | 75.30 | **77.37** | **74.37** | **46.08** | **66.16** | **45.42** |
| 50% | TEAL | 65.27 | 77.53 | 41.20 | 71.39 | 73.33 | 71.17 | 42.66 | 63.22 | 39.57 |
| | LaRoSA | 67.32 | 77.69 | **43.40** | 74.37 | 73.85 | 72.31 | 43.34 | 64.61 | 43.10 |
| | NorSA | **68.67** | **78.29** | **43.40** | **74.86** | **75.29** | **73.36** | **44.54** | **65.49** | **43.25** |
| 60% | NorSA | 67.01 | 77.86 | 42.40 | 73.30 | 72.57 | 71.59 | 42.41 | 63.88 | 39.18 |

Table 8: Performance comparison of different sparsification methods on LLaMA-2-70B.

| Sparsity | Method | WinoGrande | PiQA | OBQA | HellaSwag | BoolQ | ARC-E | ARC-C | Avg | MMLU |
|---|---|---|---|---|---|---|---|---|---|---|
| 0% | Dense | 77.97 | 82.80 | 48.80 | 83.81 | 83.79 | 81.10 | 57.33 | 73.66 | 68.80 |
| 25% | CATS | 76.32 | 82.26 | **49.20** | **84.22** | 79.94 | 79.63 | **57.59** | 72.74 | 67.50 |
| | TEAL | **78.56** | 82.42 | 48.80 | 82.66 | 83.25 | 81.26 | 56.24 | 73.31 | 67.90 |
| | LaRoSA | 77.58 | 80.09 | 48.60 | 83.63 | **83.88** | **82.70** | 57.17 | 73.38 | **68.74** |
| | NorSA | 78.22 | **82.97** | **49.20** | 83.70 | 83.70 | 80.56 | **57.59** | **73.71** | 68.56 |
| 40% | CATS | 67.25 | 77.69 | 43.00 | 75.11 | 71.41 | 61.11 | 43.60 | 62.74 | 55.83 |
| | TEAL | 76.21 | 81.65 | 48.30 | 82.02 | 82.94 | **80.87** | 55.30 | 72.47 | 66.78 |
| | LaRoSA | 77.43 | **83.57** | 47.40 | 83.37 | **83.36** | 80.51 | 57.76 | 73.31 | 68.16 |
| | NorSA | **78.14** | 82.59 | **49.00** | 83.58 | 83.30 | 80.68 | **58.19** | **73.64** | **68.22** |
| 50% | TEAL | 75.74 | 81.33 | 47.40 | 80.67 | 81.80 | **80.35** | 56.16 | 71.92 | 64.43 |
| | LaRoSA | 76.80 | **82.86** | **48.80** | 82.99 | **82.78** | 79.63 | 56.14 | 72.86 | 67.57 |
| | NorSA | **77.98** | **82.86** | **48.80** | 83.51 | 82.48 | 80.60 | **56.91** | **73.31** | **67.73** |
| 60% | NorSA | 76.95 | 82.37 | 49.40 | 82.80 | 80.76 | 80.35 | 55.80 | 72.63 | 66.75 |

Table 9: Performance comparison of different sparsification methods on LLaMA-3-8B.

| Sparsity | Method | WinoGrande | PiQA | OBQA | HellaSwag | BoolQ | ARC-E | ARC-C | Avg | MMLU |
|---|---|---|---|---|---|---|---|---|---|---|
| 0% | Dense | 73.64 | 80.41 | 44.80 | 79.13 | 81.10 | 77.78 | 53.50 | 70.05 | 65.26 |
| 25% | CATS | 70.24 | 79.33 | 44.40 | 77.38 | 79.17 | 73.70 | 49.40 | 67.66 | 61.85 |
| | TEAL | 72.48 | 80.36 | **45.40** | 78.45 | 80.07 | 77.08 | 52.00 | 69.40 | 63.85 |
| | LaRoSA | **72.93** | **80.63** | 45.00 | **79.09** | 80.40 | 76.68 | 52.05 | 69.54 | 64.85 |
| | NorSA | **72.93** | 80.52 | 45.00 | 79.03 | **81.22** | **77.53** | **53.16** | **69.91** | **65.06** |
| 40% | CATS | 60.54 | 73.12 | 36.00 | 66.82 | 59.14 | 52.78 | 37.37 | 55.11 | 31.82 |
| | TEAL | 70.34 | 78.51 | **45.80** | 76.55 | 79.57 | 76.47 | 49.74 | 68.14 | 59.84 |
| | LaRoSA | 71.11 | 79.87 | 44.00 | 77.85 | 80.09 | 76.98 | 51.62 | 68.79 | 62.61 |
| | NorSA | **73.01** | **81.07** | 44.80 | **78.56** | **80.43** | **77.74** | **52.39** | **69.71** | **64.07** |
| 50% | TEAL | 66.71 | 77.09 | 42.80 | 72.61 | 76.26 | 72.71 | 46.24 | 64.92 | 52.78 |
| | LaRoSA | 68.98 | 79.98 | 44.20 | 76.85 | 77.55 | 74.75 | 48.04 | 67.19 | 58.65 |
| | NorSA | **72.14** | **80.09** | **44.40** | **77.70** | **79.60** | **76.60** | **51.19** | **68.82** | **62.12** |
| 60% | NorSA | 70.80 | 78.73 | 43.40 | 75.84 | 78.26 | 75.63 | 48.63 | 67.33 | 57.96 |

Table 10: Performance comparison of different sparsification methods on LLaMA-3-70B.

| Sparsity | Method | WinoGrande | PiQA | OBQA | HellaSwag | BoolQ | ARC-E | ARC-C | Avg | MMLU |
|---|---|---|---|---|---|---|---|---|---|---|
| 0% | Dense | 80.11 | 84.49 | 48.80 | 84.97 | 85.22 | 86.15 | 64.33 | 76.29 | 78.71 |
| 25% | CATS | 79.87 | 84.06 | 47.20 | **85.21** | 84.43 | 85.19 | 63.14 | 75.58 | 77.93 |
| | TEAL | 76.63 | 82.31 | 48.20 | 81.82 | 83.24 | 81.64 | 58.78 | 73.23 | 74.86 |
| | LaRoSA | **81.45** | 84.39 | **49.00** | 85.07 | 84.89 | 85.65 | 63.65 | **76.30** | 78.13 |
| | NorSA | 80.27 | **84.66** | 48.40 | 84.97 | **85.26** | **86.15** | **64.25** | 76.28 | **78.71** |
| 40% | CATS | 74.51 | 81.77 | 46.40 | 83.46 | 81.44 | 79.17 | 57.76 | 72.07 | 72.12 |
| | TEAL | 76.32 | 81.93 | 45.80 | 81.56 | 80.64 | 81.90 | 57.50 | 72.24 | 73.23 |
| | LaRoSA | 79.72 | 83.62 | **48.40** | 84.71 | 85.05 | 84.26 | 62.12 | 75.41 | 77.62 |
| | NorSA | 79.48 | **84.39** | 47.80 | **84.80** | **85.08** | **85.44** | **63.99** | **75.85** | **78.29** |
| 50% | TEAL | 73.08 | 80.30 | 43.60 | 79.90 | 81.74 | 80.17 | 56.82 | 70.80 | 69.20 |
| | LaRoSA | 77.90 | 82.59 | 46.40 | 84.24 | 84.10 | 82.49 | 58.96 | 73.81 | 76.51 |
| | NorSA | **78.85** | **83.62** | **47.20** | **84.56** | **84.86** | **85.23** | **62.37** | **75.34** | **78.15** |
| 60% | NorSA | 78.22 | 83.19 | 47.60 | 84.14 | 84.86 | 84.51 | 62.37 | 74.98 | 76.09 |

Table 11: Performance comparison of different sparsification methods on Mistral-7B-v0.3.

| Sparsity | Method | WinoGrande | PiQA | OBQA | HellaSwag | BoolQ | ARC-E | ARC-C | Avg | MMLU |
|---|---|---|---|---|---|---|---|---|---|---|
| 0% | Dense | 73.80 | 82.31 | 44.00 | 80.40 | 82.11 | 78.40 | 52.04 | 70.44 | 62.34 |
| 25% | CATS | 71.27 | 80.47 | 33.00 | 60.68 | 79.42 | 78.28 | 48.72 | 64.55 | 59.83 |
| | TEAL | 72.61 | 81.82 | **44.40** | 80.05 | 81.44 | 78.24 | 51.86 | 70.06 | 61.51 |
| | LaRoSA | 72.85 | **82.15** | **44.40** | 80.14 | 81.71 | **78.49** | 52.05 | 70.25 | 61.81 |
| | NorSA | **73.88** | **82.15** | 44.20 | **80.29** | **82.29** | 78.45 | **52.22** | **70.49** | **62.22** |
| 40% | CATS | 61.96 | 74.43 | 34.80 | 74.71 | 74.86 | 55.13 | 41.47 | 59.62 | 44.31 |
| | TEAL | 69.55 | 80.36 | 44.40 | 79.08 | 80.49 | 77.14 | 50.30 | 68.76 | 60.17 |
| | LaRoSA | 70.88 | 81.07 | 43.40 | 79.63 | **81.68** | **78.32** | 51.11 | 69.44 | 61.15 |
| | NorSA | **72.77** | **81.77** | **44.80** | **80.16** | 81.56 | 78.16 | **52.82** | **70.29** | **61.23** |
| 50% | TEAL | 68.71 | 79.52 | 40.40 | 76.97 | 79.55 | 74.34 | 47.63 | 66.73 | 57.34 |
| | LaRoSA | 69.93 | **81.61** | 43.20 | 78.93 | 80.92 | 75.55 | 49.06 | 68.46 | 58.80 |
| | NorSA | **70.56** | **81.61** | **45.00** | **80.03** | **81.56** | **78.07** | **52.05** | **69.84** | **60.41** |
| 60% | NorSA | 71.35 | 81.07 | 43.60 | 79.00 | 80.83 | 77.31 | 49.66 | 68.97 | 58.63 |

Table 12: Performance comparison of different sparsification methods on Qwen-2.5-7B.

| Sparsity | Method | WinoGrande | PiQA | OBQA | HellaSwag | BoolQ | ARC-E | ARC-C | Avg | MMLU |
|---|---|---|---|---|---|---|---|---|---|---|
| 0% | Dense | 72.77 | 79.98 | 47.60 | 78.91 | 84.56 | 77.44 | 51.11 | 70.34 | 74.21 |
| 25% | CATS | 70.64 | 79.54 | 46.20 | **78.90** | 83.15 | **78.28** | 50.94 | 69.66 | 72.67 |
| | TEAL | 72.12 | 79.22 | 46.40 | 78.49 | 84.40 | 76.85 | 50.85 | 69.76 | 73.21 |
| | LaRoSA | **72.69** | **79.92** | 46.60 | 78.77 | 84.40 | 77.02 | **51.45** | **70.12** | 73.74 |
| | NorSA | 71.67 | 79.82 | **47.00** | 78.72 | **84.65** | 76.94 | 51.02 | 69.97 | **73.90** |
| 40% | CATS | 58.72 | 76.12 | 38.60 | 72.49 | 74.01 | 67.89 | 44.97 | 61.83 | 63.99 |
| | TEAL | 70.30 | 78.78 | 45.00 | 76.78 | 84.05 | 75.44 | 49.96 | 68.61 | 71.44 |
| | LaRoSA | 70.72 | 79.38 | 46.20 | 78.28 | **84.65** | **77.86** | 50.60 | 69.67 | 72.33 |
| | NorSA | **71.35** | **79.82** | **46.40** | **78.43** | 84.31 | 76.98 | **50.77** | **69.72** | **73.35** |
| 50% | TEAL | 68.90 | 77.18 | 44.00 | 74.60 | 83.76 | 76.59 | 49.34 | 67.76 | 68.53 |
| | LaRoSA | **70.24** | 79.11 | **45.60** | 77.20 | 83.55 | **78.03** | 49.91 | 69.09 | 70.09 |
| | NorSA | 69.85 | **79.54** | **45.60** | **77.59** | **84.31** | 76.56 | **51.37** | **69.26** | **72.28** |
| 60% | NorSA | 70.01 | 79.00 | 43.80 | 75.75 | 84.04 | 77.19 | 50.94 | 68.67 | 69.78 |

Table 13: Performance comparison of different sparsification methods on Qwen-2.5-72B.

| Sparsity | Method | WinoGrande | PiQA | OBQA | HellaSwag | BoolQ | ARC-E | ARC-C | Avg | MMLU |
|---|---|---|---|---|---|---|---|---|---|---|
| 0% | Dense | 77.90 | 83.62 | 46.40 | 86.05 | 89.14 | 83.33 | 62.62 | 75.58 | 86.08 |
| 25% | CATS | 73.72 | 82.86 | 45.40 | **86.60** | 82.57 | 79.92 | 58.36 | 72.77 | 84.91 |
| | TEAL | **78.11** | 83.39 | 46.50 | 85.23 | 87.63 | 82.62 | 61.87 | 75.05 | 85.44 |
| | LaRoSA | 77.66 | 83.73 | **46.80** | 85.97 | **89.17** | 82.83 | **62.54** | 75.53 | 85.62 |
| | NorSA | 78.06 | **84.22** | 46.60 | 85.93 | 88.90 | **83.04** | 62.29 | **75.57** | **85.80** |
| 40% | CATS | 65.59 | 78.56 | 33.20 | 61.81 | 82.14 | 74.71 | 45.82 | 63.12 | 80.95 |
| | TEAL | 77.47 | 83.01 | 46.50 | 84.52 | 87.52 | 82.43 | 61.11 | 74.65 | 84.80 |
| | LaRoSA | 78.45 | 83.03 | **47.20** | **86.01** | 89.48 | 83.12 | **62.46** | 75.35 | 85.33 |
| | NorSA | **79.32** | **83.73** | 47.00 | 85.88 | 88.81 | **83.38** | 61.43 | **75.65** | **85.83** |
| 50% | TEAL | 77.32 | 82.25 | 45.00 | 83.03 | 87.54 | 81.61 | 59.46 | 73.74 | 83.54 |
| | LaRoSA | 76.48 | 83.57 | 45.60 | 85.81 | **89.51** | **83.59** | **61.69** | 75.18 | 84.34 |
| | NorSA | **77.51** | **83.68** | **47.80** | **85.85** | 88.81 | 83.08 | 61.43 | **75.45** | **85.66** |
| 60% | NorSA | 77.19 | 83.13 | 47.20 | 85.53 | 88.59 | 82.24 | 61.95 | 75.12 | 84.96 |