# OpenReview forum: "NorSA: Accelerate LLM Decoding via Normalized Sparse Activation"
_ICLR.cc/2026/Conference — Submitted to ICLR 2026_

### Official Review · Reviewer_pMJ8 · 2025-10-17

**Soundness:** 3
**Presentation:** 2
**Contribution:** 2
**Rating:** 4
**Confidence:** 3

**Summary:**

The study introduces Normalized Sparse Activation (NorSA), a method to speed up large language model decoding by reducing some of the unnecessary computations. The key insight of the method, different to others, is that it correctly does not assume that the generated tokens are IID, but they are casual (a token is conditioned in all the previous generated tokens). Therefore, the method does a simple adjustement when choosing the tokens to threshold. Furthermore, it also applies normalization, rotation, and thresholding to maintain meaningful activations. The method is tested in several benchmarks and using 3 family of LLMs (LLaMA, Mistral and Qwen) reaching results that outperform the other two methods it compares with.

**Strengths:**

I think these are the main strengths of the paper:

1) The main part of the method (normalized sparse attention) is well-motivated, makes very much sense, and is very simple to understand. This is actually quite nice, considering that most of the papers try to over-complicate things in order to make them sound novel.

2) The results are quite good. The method decisively outperforms TEAL and La RoSA in several benchmarks.

3) There are some interesting ablation studies shown in the method.

4) Very nice to see that the authors also do a hardware aware kernel for the method.

**Weaknesses:**

I think these parts of the paper can be further improved:

1) It is unclear to me the connection between sections 4.2 and 4.3. In particular, while I really like the motivation of 4.2, to some degree, 4.3 looks to me a bit forced, almost like trying to increase the complexity of the method. Furthermore, I think there must be an ablation that shows the performance of both 4.2 and 4.3 in isolation, without them being combined.

2) Presentation

2a) Related work can be further improved. Right now, it just mentions some papers, but without clarifying how do they work, why they are important and how do they connect with this work.

2b) Figures 1 and 2 can be massively improved. There is a lot of trivial information there (the attention mechanism) that can be collapsed.

2c) Table 2 comes before Table 1 in the paper, this should be rearranged.

**Questions:**

I would appreciate if the authors can clarify these potential issues:

1) Why the timing performance has been done in A100 GPUs? Is it just because that is what the authors have or some other reasons? It would be ideal if we could see some results in more modern H100 (or B-series) GPUs.

2) What does 100% sparsity even mean?

3) Are all the results shown while also having standard speedup mechanisms such as FlashAttention and KV caching?

I am quite willing to increase my score based on the answers for Weaknesses and Questions.

---

> ### Author Response · Authors · 2025-11-17
> **Response to Reviewer pMJ8's Comments (1/1)**
>
> Thank you for your appreciation of our work! We sincerely appreciate the time and effort you have dedicated to reviewing our paper and providing valuable feedback. We are grateful for the opportunity to address your concerns and offer further clarification.
>
>
> **Weakness 1: Connection Between Sections 4.2 and 4.3**
>
> The introduction of rotation matrices in Section 4.3 aims to further enhance the sparsification function proposed in Section 4.2 by reducing correlations between different dimensions.
>
> For detailed **theoretical analysis** and **ablation study** of Section 4.3, please refer to our **main rebuttal**.
>
> From the experimental results in the main rebuttal, the following observations can be made:
>
> 1. In almost all sparsity levels and models, NorSA w/o R performs better than TEAL, indicating the effectiveness of Section 4.2. Using normalized hidden states for thresholding offers better performance compared to directly thresholding based on absolute values.
>
> 2. In almost all sparsity levels and models, NorSA (combination of Sections 4.2 and 4.3) outperforms NorSA w/o R (only Section 4.2), demonstrating the effectiveness of Section 4.3, with the margin of improvement increasing at higher sparsity levels.
>
> 3. NorSA (combination of normalized sparsify from Section 4.2 and rotations from Section 4.3) outperforms LaRoSA (combination of top-K sparsify and rotations), indicating that the normalized sparsify proposed in Section 4.2 is more effective, with smaller computational complexity.
>
>
>
> **Weakness 2: Presentation**
>
> (a) Thank you for your suggestion! We have revised the related work section to provide a clearer understanding of how past studies operate, their significance, and their connection to our work. Through our adjustments, we aim to present a comprehensive view of the landscape, illustrating where our approach fits in and how it seeks to push the boundaries established by previous efforts.
>
> (b) Thank you for your suggestion, and we truly value your feedback. However, because the rotation matrices $R_2$ and $R_2^T$, as proposed in Section 4.3, need to be integrated into the V and O matrices of the attention mechanism, it is challenging for us to further simplify the current diagrams without losing clarity. These details are essential for fully illustrating our methodology of Section 4.3. If you have specific suggestions for improvement, we would greatly appreciate them and consider implementing changes accordingly!
>
> (c) Yes, this was an oversight. We have rearranged the tables in the latest version of the paper. The table numbers now follow a logical order consistent with the reading flow.
>
> **Question 1: Device**
> Our experiments were conducted using A100 GPUs due to the resources we currently have available. We recognize the interest in seeing results on more modern GPUs like H100 or B-series, as these would offer insights into performance on the latest hardware. Unfortunately, we do not have access to these newer GPUs at the moment. Thank you for your understanding.
>
> **Question 2: 100% Sparsity**
> The kernel's runtime is composed of various components, including kernel launch, data loading, computations, and more. In this paper, the Triton implementation we provided targets acceleration primarily in the weight reading component, which scales linearly with sparsity. Setting sparsity to 100% implies the complete elimination of weight reading, causing the runtime to reflect only the portions that cannot be accelerated.
>
> According to Amdahl's Law, the potential speedup of a system is limited by the proportion of the system that can be parallelized or accelerated. By examining the runtime at 100% sparsity, we gain insights into the limits of acceleration. This provides valuable perspectives for more detailed future optimization strategies, identifying areas that could benefit from further refinement and acceleration.
>
> **Question 3: FlashAttention and KV caching**
> FlashAttention was not used in our experiments. However, we did utilize KV caching to enhance performance. We have not tested our method in long context scenarios, where latency is mainly determined by prefill and attention operations, which have specialized optimization techniques. Our method is orthogonal to these techniques and can be used in conjunction with them.
>
>
>
> For your questions, we have carefully prepared thorough responses. If you have any further inquiries, we are more than happy to address them. We hope our answers have effectively resolved your concerns, and we invite you to consider raising your score based on our clarifications.
>
> Thank you for your thoughtful review!

---

### Official Review · Reviewer_mCoY · 2025-10-31

**Soundness:** 2
**Presentation:** 1
**Contribution:** 2
**Rating:** 4
**Confidence:** 5

**Summary:**

The authors present that existing activation sparsification techniques (e.g., TEAL, LaRoSA) rely on the flawed assumption of i.i.d. activations. To address this, NorSA introduces norm-based thresholding (to incorporate contextual scale information) and rotation matrices (to decorrelate activation dimensions). Experiments across LLaMA, Mistral, and Qwen families show the efficacy of the approach.

**Strengths:**

- The motivation of iid assumption not holding is reasonable.
- Having numerical results across different model families and kernel implementation is good.

**Weaknesses:**

- Lack of novelty: The rotation idea is adopted from SliceGPT paper.

- Lack of clarity. The context-aware selection seems equivalent to the layer-wise sparsity allocation. In other words, though other works select activated neurons upon some threshold, the thresholds could be assigned in the global cross-layer information, making them context-aware as well.

- The writing needs to be improved. There are many inconsistencies of writing format throughout the paper.

**Questions:**

See the weakness.

---

> ### Author Response · Authors · 2025-11-17
> **Response to Reviewer mCoY's Comments (1/2)**
>
> Thank you for your detailed review and feedback on our submission. We sincerely appreciate the time and effort you have dedicated to reviewing our paper and providing valuable feedback. We are grateful for the opportunity to address your concerns and offer further clarification.
>
>
> **Weakness 1: Addressing the Novelty Concern**
>
> The primary focus and contribution of our paper is to reveal the limitations of the independent and identically distributed (i.i.d.) assumption on which previous sparse activation methods are based, and to propose effective compensatory approaches that are validated by both theoretical and empirical evidence: normalized sparse activation and correlation-reducing rotations.
>
> Our approach differs from SliceGPT, which focus on static weight pruning (i.e., the parameters used to decode each token are the same, and the goal is to optimize the expected performance of the model over a set of tokens). In contrast, we focus on dynamic sparse activation, where the parameters used to decode each token can change dynamically, aiming to optimize the model's performance for every individual token. This difference in objective also leads to different theoretical analyses. SliceGPT uses rotation with the motivation to reconstruct input in the optimal way, whereas we use rotation with the motivation to improve the accuracy of estimating dimension importance (please refer to our main rebuttal for more details). Furthermore, our method is applicable to scenarios where multiple hidden states/activations share one rotation matrix, broadening its usability across varying model configurations.
>
> Additionally, we provide both theoretical insights and empirical evidence on which types of rotation matrices can enhance sparse activation performance. A crucial point is that not all rotation matrices can improve sparse activation performance; some can even be detrimental. As illustrated in Sections 4.2 and 5.2, the model's performance varies depending on the rotation matrices generated with different optimization steps, showing that some matrices can enhance performance while others may detract from it. This highlights the importance of how to derive rotation matrices that lead to performance improvements. Our work contributes by providing the method for obtaining such rotation matrices and offering theoretical support that rotation matrices reducing inter-dimensional correlations can enhance the performance of models with sparse activation, thereby inspiring future research in this area.

---

> ### Author Response · Authors · 2025-11-17
> **Response to Reviewer mCoY's Comments (2/2)**
>
> **Weakness 2: Addressing Clarity and Contextual Understanding**
>
> Thank you for your suggestion, which provides a valuable insight into broader exploration.
>
> Our work focuses on capturing the dependencies and context within the data that the model receives. We acknowledge that the term "context" may not have been clearly defined in our paper, potentially leading to misunderstandings. In our work, "context" specifically refers to the token sequences received or generated by the model, with our modeling objective focusing on the dependencies between individual tokens within these sequences. These dependencies are encoded within the activation/hidden states, which are represented as high-dimensional vectors composed of values across various dimensions, where there exist potential inter-dimensional correlations.
>
> Compared to the magnitude-based threshold methods used by TEAL [1], NorSA exhibits a stronger capability to capture contextual information. In magnitude-based threshold methods, for an activation/hidden state $X \in \mathbb{R}^{d_{hidden}}$, the decision to set each dimension's value to zero is quasi independent (i.e., the events of $X_i$ being removed and $X_j$ being removed are quasi independent since each dimension is compared to the same predetermined constant value $\tau$):
>
> $$
> \text{Sparsify}_{\text{TEAL}}(X)_i = \begin{cases}
> X_i & \text{if } |X_i| > \tau \\
> 0 & \text{otherwise}
> \end{cases}
> $$
>
> In contrast, NorSA's normalization approach makes the removal of a dimension dependent on the values of all other dimensions, as the threshold is determined based on the current activation/hidden state $X \in \mathbb{R}^{d_{hidden}}$ received, and the norm squared is the sum of squares of each dimension:
>
> $$
> \text{Sparsify}_{\text{NorSA}}(X)_i = \begin{cases}
> X_i & \text{if } |X_i| > \tau \cdot \|X\| \\
> 0 & \text{otherwise}
> \end{cases}
> $$
>
> As a result, NorSA considers more information, thereby becoming more context-aware.
>
> Layer-wise sparsity takes into account the relationships between various components within the model. It allocates sparsity levels to each layer by considering the overall structure and weights of the model. In the context of layer-wise sparsity, "context" refers to the relationships between various components within the model, which is information about the model itself rather than the data.
>
> Your opinion provides a great direction for exploration. The information about the model can be combined with the information about the data. Our proposed method could be combined with layer-wise sparsity strategies. In fact, we already used layer-wise sparsity in our paper. In the paragraph of Section 5 on "Mixed Sparsity Configuration," we describe our use of the block-wise greedy optimization technique proposed by TEAL [1] to configure mixed sparsity within a single transformer layer. Each transformer layer has the same total sparsity, but the sparsity of the individual components (e.g., Q, K, V, O) within each transformer layer differs. In the future, further layer-wise sparsity can be explored, such as assigning different sparsity levels to each transformer layer. However, this direction is beyond the scope of this paper and we leave it for future work.
>
> [1] Training-Free Activation Sparsity in Large Language Models
>
> **Weakness 3: Writing and Format Consistency**
>
> Thank you for your thorough reading of our paper. We have already addressed, and will continue to address, the inconsistencies in writing format throughout the paper.  We would greatly appreciate if you could provide more specific examples or areas where improvements are needed, as it will help us enhance the clarity and coherence of our presentation.
>
>
> Thank you once again for your constructive feedback. If you have any further inquiries, we are more than happy to address them. We hope our answers have effectively resolved your concerns, and we invite you to consider raising your score based on our clarifications.

---

### Official Review · Reviewer_L5Vk · 2025-11-01

**Soundness:** 3
**Presentation:** 3
**Contribution:** 3
**Rating:** 6
**Confidence:** 3

**Summary:**

This work introduces Normalized Sparse Activation (NorSA) and enhances prior approaches by considering the correlations between the dimensions of hidden states.

**Strengths:**

1. This work challenges the assumption that different activation dimensions are independent and suggests a way to incorporate that.
2. Combining the norm calculation with other operations is an effective way to reduce overhead.
3. The author invested considerable effort into the implementation and achieved impressive empirical results across different scales.

**Weaknesses:**

Some of the technical and experimental designs might need additional motivation; please see "Questions".

**Questions:**

1. I have some reservations about Equation (3). The authors compare activation entries to their norm, but these seem like different quantities, making the comparison questionable. Why not use $\tau$ standard deviations from the mean instead?

2. Regarding equations (5) and (6), is the PCA-based rotation matrix an approximate solution? Also, I'm unsure why a rotation matrix would be useful here, as I usually think of it for smoothing outliers.

3. In Section 5.2, how do the authors "learn" the rotational matrix?

4. Is there a typo after Equation (4)? It references `sparsify` twice.

5. I haven't seen any speed comparisons with other methods. Is the main focus that the improvements are mainly in quality with minimal latency impact? If so, that seems reasonable to me.

---

> ### Author Response · Authors · 2025-11-17
> **Response to Reviewer L5Vk's Comments (1/2)**
>
> Thank you for your detailed review and insightful comments on our submission! We sincerely appreciate the time and effort you have dedicated to reviewing our paper and providing valuable feedback. We are committed to addressing your questions and concerns to the best of our ability.
>
> **Question 1: Concerns about Equation (3) and Comparison Discussion**
>
> Equation (3) is designed to compare each element $X_i \in \mathbb{R}$ of a high-dimensional vector $X \in \mathbb{R}^{d_{hidden}}$ to the norm $||X||$ of this vector, scaled by a priori coefficient $\tau$. In this context, $X$ represents the high-dimensional vector, with $X_i$ as the $i^{th}$ component.
>
> To draw an analogy with signal processing, particularly involving Fourier transforms, consider $X \in \mathbb{R}^{d_{hidden}}$ as a composite signal composed of $d_{hidden}$ different frequency sine waves. Here, $X_i$ is the amplitude of the sine wave signal at the $i^{th}$ frequency, and $|X_i|^2$ indicates the energy contributed by this frequency. According to Parseval's theorem, the squared norm $||X||^2 = \sum_j |X_j|^2$ is the sum of energies across all these frequency sine waves. Therefore, Equation (3) selectively filters out frequencies with lower energy contributions, retaining those with higher contributions. As such, $|X_i|$ and $||X||$ are fundamentally similar quantities, making their comparison logical. The coefficient $\tau$ establishes the threshold for this comparison.
>
> The choice of $\tau$ is guided by the formula $\mathbb{E}_{X \sim p_{\text{data}}}[\text{Sparsity}(\text{Sparsify}_{\text{NorSA}}(X))] = \text{target sparsity}$, ensuring that the expected sparsity level matches the desired target sparsity.
>
> We must admit that we did not fully understand the suggestion "use $\tau$ standard deviations from the mean instead", and we would appreciate further clarification on this aspect.
>
> **Question 2: Rotation Matrix and Its Role**
>
> Yes, the PCA-based rotation matrix indeed serves as an approximate solution for Equations (5) and (6).
>
> We present the impact of rotation matrices from both **theoretical** and **empirical perspectives** in main rebuttal and strongly recommend you to read them.
>
> One interesting thing to note is that the capabilities of rotation matrices extend beyond smoothing outliers. They can also unsmooth ouliers! Rotation or orthogonal matrices belong to a class of matrices satisfying the properties $R^T R = I$ and $R^T = R^{-1}$. An intriguing aspect is that if a rotation matrix $R$ can smooth outliers in weight $W$ (expressed as $W' = WR$), its transpose $R^T$ (also a rotation matrix) can reverse this smoothing ($W'R^T = WRR^T = W$). Thus, rotation matrices can also unsmooth outliers, and do something else.

---

> ### Author Response · Authors · 2025-11-17
> **Response to Reviewer L5Vk's Comments (2/2)**
>
> **Question 3: Learning the Rotational Matrix**
>
> We detail the process of learning rotational matrices through an optimization framework that leverages gradient descent. The learning process is designed to iteratively refine the rotational matrices to enhance sparse activation performance. Here is the stepwise approach:
>
> 1. **Merge RMSNorm Weights**: Initially, the RMSNorm weights are merged into the subsequent weight matrices to streamline computations.
>
> 2. **Forward Propagation**: Perform a forward pass on a calibration dataset, capturing the hidden states/activations received by each weight matrix.
>
> 3. **PCA-derived Matrix Calculation (Optional)**: Optionally, calculate PCA-derived matrices based on the captured hidden states to provide a strong initialization point.
>
> 4. **Integrate Rotational Matrices**: Integrate rotational matrices initialized either as Identity matrices or PCA-derived matrices to begin the refinement process.
>
> 5. **Optimize Rotational Matrices**: These matrices are then optimized using Equations (5) and (6) as loss functions, where both weights and hidden states/activations are kept frozen during optimization to focus solely on adjusting the rotations.
>
> This entire procedure can be executed layer by layer.
>
>
> **Response to Question 4: Typo After Equation (4)**
>
> We appreciate your attention to detail. We surmise that you are referring to the formula of $\tau$:
>
> $$ \mathbb{E}\_{X \sim p\_{\text{data}}}[\text{Sparsity}(\text{Sparsify}\_{\text{NorSA}}(X))] = \text{target sparsity} $$
>
> In this expression, "Sparsity()" is a function that measures the proportion of zero elements within a high-dimensional vector (mapping from $\mathbb{R}^{d_{hidden}}$ to $[0, 1]$). Meanwhile, "Sparsify()" is a function that transforms certain elements within a high-dimensional vector to zero (mapping from $\mathbb{R}^{d_{hidden}}$ to $\mathbb{R}^{d_{hidden}}$).
>
> **Response to Question 5: Speed Comparisons and Focus on Quality**
>
> Yes, our proposed method could enhance model quality with minimal latency impact. In Section 4.4 of the paper, we provide a detailed theoretical analysis of NorSA's computational complexity, highlighting that NorSA is faster than La RoSA and slightly slower than TEAL.
>
> For throughput comparisons, we invite you to contrast our reported results with those in the aforementioned papers. Do note that variations may occur due to differences in the systems and underlying hardware configurations used to run performance tests. Additionally, you may refer to the matrix multiplication acceleration data provided in Section 5.4 for further insights into the speed improvements achieved by NorSA.
>
>
> For your questions, we have carefully prepared thorough responses. If you have any further inquiries, we are more than happy to address them. We hope our answers have effectively resolved your concerns, and we invite you to consider raising your score based on our clarifications.
>
> Thank you for your thoughtful review!

---

### Official Review · Reviewer_DQe5 · 2025-11-17

**Soundness:** 3
**Presentation:** 2
**Contribution:** 2
**Rating:** 4
**Confidence:** 3

**Summary:**

NorSA is a training-free activation sparsity method for LLMs that relaxes the i.i.d. activation assumption used by TEAL / La RoSA. It uses a norm-normalized threshold (|xᵢ| > τ‖x‖) so sparsity decisions depend on the overall scale of each token’s hidden state, and introduces PCA-based rotation matrices to reduce linear correlations between dimensions before sparsification. Experiments on LLaMA-2/3, Mistral, and Qwen-2.5 show better perplexity and zero-shot / MMLU performance than TEAL and La RoSA at the same sparsity, plus real decoding speedups with Triton kernels.

**Strengths:**

- The normalized per-hidden-state thresholding rule is simple, intuitive, and easy to integrate into existing models.

- Strong empirical results across multiple model families and sparsity levels, consistently surpassing TEAL and La RoSA, especially at higher sparsity.

- Hardware-aware implementation (fused SwiGLU+norm, sparse GEMV) and ablations on rotations / covariance make the method feel practically usable.

**Weaknesses:**

- Conceptual novelty is moderate: norm-normalized thresholds + PCA rotations sit close to prior rotation-based sparsity work; the paper could be clearer about what is genuinely new.

- There is no clean, large-scale ablation of “NorSA without rotations” vs “NorSA with rotations”.

- The choice of PCA as the rotation mechanism is under-motivated: no comparison to learned rotations (e.g., via distillation), and limited discussion of calibration cost and scaling.

**Questions:**

- PCA vs learned rotations: Why did you choose PCA-based rotations over learning rotations with a small teacher–student distillation objective or gradient-based optimization?
- TEAL and the i.i.d. assumption: You argue that prior work (e.g., TEAL) implicitly assumes i.i.d. activations across dimensions. Could you make this more explicit? What part of TEAL’s design depends on that assumption, and can you show empirical deviations from i.i.d. in real hidden states?

---

> ### Author Response · Authors · 2025-11-20
> **Response to Reviewer DQe5's Comments (1/4)**
>
> Thank you for your detailed review and insightful comments on our submission! We sincerely appreciate the time and effort you have dedicated to reviewing our paper and providing valuable feedback. We are committed to addressing your questions and concerns to the best of our ability.
>
>
> **Question 2: The i.i.d. Assumption and TEAL**
>
> The i.i.d. assumption is explicitly mentioned in the motivation section of TEAL's paper. [1]
>
> In TEAL's paper, Section 4.1, "MOTIVATING STUDY: DISTRIBUTIONAL PROPERTIES OF ACTIVATIONS IN LLMS," outlines some general observations regarding activation distributions. To quote from TEAL's paper: "LLM weights are typically Gaussian, and multiplying an independent isotropic Gaussian vector with an independent Gaussian matrix follows a multivariate generalized Laplace distribution (the weights and activations are clearly not independent in practice). Attention is a data-dependent linear operator which may have similar properties. Distributions may be zero-mean due to layer normalization.”
>
> Now, let's analyze the sparsification function proposed by TEAL, which prunes low-magnitude activations:
>
> $$
> \text{Sparsify}\_{\text{TEAL}}(X)\_i = \begin{cases}
> X\_i & \text{if} |X\_i| > \tau  \\
> 0 & \text{otherwise}
> \end{cases}
> $$
>
> It has the following characteristics:
>
> 1. The core of the sparsification function proposed by TEAL: the threshold $\tau$ is a constant and does not change with activation/hidden state, meaning each activation/hidden state is sparsified by the identical function. The events of sparsification for any two distinct activation/hidden states are independent.
>
> 2. Every dimension within every activation/hidden state is compared against the same constant threshold $\tau$, indicating the events of removal for any two different dimensions within the same activation/hidden state or accross different activation/hidden state are independent.
>
> We think that if the assumption that each activation/hidden state is a realization of an i.i.d. high-dimensional random vector holds, then TEAL's design of the sparsification function is reasonable, as the properties of independence and identity are maintained. However, if this assumption does not hold, TEAL's sparsification function may loss significant information due to its simplicity. Therefore, we argue that TEAL implicitly relies on the i.i.d. assumption.
>
> Additionally, in Appendix A.1, TEAL's authors derive the expected relative error induced by $ \text{Sparsify}\_{\text{TEAL}} $  under a restrictive assumption that weights and activations are independent Gaussians.
>
> In Figure 10 of Appendix A.1, the TEAL authors compare theoretical expected relative errors based on i.i.d. Gaussian assumptions with empirical relative errors of real hidden states induced by $ Sparsify_{TEAL} $. The results show that the theoretical expected relative errors from the i.i.d. Gaussian are larger than the actual relative errors. For example, at 40% sparsity for $W\_{down}$, the theoretical relative error is about 20%, while the actual relative error is only about 10%. This comparison shows the discrepancy between theoretical assumptions and empirical observations, implying that the i.i.d. assumption does not hold perfectly in practical scenarios.
>
> [1] Training-Free Activation Sparsity in Large Language Models

---

> ### Author Response · Authors · 2025-11-20
> **Response to Reviewer DQe5's Comments (2/4)**
>
> **Weakness 3 and Question 1: Rotation**
>
> **Motivation**
>
> Our core purpose is reducing correlations between different dimensions of activation/hidden states. This motivation is driven by two objectives:
>
> 1. To enhance the accuracy of dimension importance estimation of $\text{Sparsify}\_{\text{NorSA}}$ proposed in Section 4.2.
> 2. To attempt reducing the dimensionality of activations/hidden states. (Naturally introduction activation sparsity.)
>
> Rotation is one method to achieve this goal, but it is not the only one.
>
> We strongly recommend reading the **detailed theoretical derivations** and **extensive empirical verification** in our **main rebuttal**.
>
> **Gradient Based Optimization**
>
> To find rotation matrices that effectively reduce inter-dimensional correlations, we train the rotation matrices using gradient descent with Eq.6 or Eq.7 from Section 4.3 as the loss function. We have already reported the use of gradient-based methods to optimize rotation matrices in the paper!
>
> The detailed training procedure is as follows:
>
> 1. **Merge RMSNorm Weights**: Initially, the RMSNorm weights are merged into the subsequent weight matrices to streamline computations.
>
> 2. **Forward Propagation**: Perform a forward pass on a calibration dataset, capturing the hidden states/activations received by each weight matrix.
>
> 3. **PCA-derived Matrix Calculation (Optional)**: Optionally, calculate PCA-derived matrices based on the captured hidden states to provide a strong initialization point.
>
> 4. **Integrate Rotational Matrices**: Integrate rotational matrices initialized either as Identity matrices or PCA-derived matrices to begin the refinement process.
>
> 5. **Optimize Rotational Matrices**: These matrices are then optimized using Equations (6) or (7) as loss functions, where both weights and hidden states/activations are kept frozen during optimization to focus solely on adjusting the rotations.
>
> This entire procedure can be executed layer by layer.
>
> We include PCA as an optional part of the above training rotation matrix process because PCA can quickly and efficiently provide an approximate optimal solution for Eq.6 or Eq.7. As Table 2 in Section 5.2 indicates, the self-linear-correlation ratio of activation/hidden states after PCA-derived rotation matrix transformation approaches the upper limit (100%). While further training can indeed improve the self-linear-correlation ratio and model performance, the improvements are marginal.
>
> **Cost Analysis**
>
> Since rotation matrices are constant, they can be factored out of the expectation, allowing Eq. 6 to be reformulated as:
>
> $$\min\_{R} \sum\_{i \neq j} \mathbb{E}\_{X \sim p\_{\text{data}}}[(R^T X^T X R)^2\_{i,j}] = \min_{R} \sum\_{i \neq j} (R^T \mathbb{E}\_{X \sim p\_{\text{data}}}[X^T X]R)^2\_{i,j}$$
>
> During optimization, the activations/hidden states are kept frozen, so the covariance matrix
> $\mathbb{E}\_{X \sim p\_{\text{data}}}[(X^T X )]$
> needs to be computed only once, at the beginning of training. It can be computed in an online fashion during the forward pass on the calibration data with a spatial complexity of
> $O(d\_{\text{hidden}}^2)$
> and a time complexity of
> $O(\text{number of calibration data} \times d\_{\text{hidden}}^2)$.
> The optional PCA has a time complexity of $O(d\_{\text{hidden}}^3)$.
>
> Subsequent gradient-based optimizations of the rotation matrix do not involve repeated computations of the covariance matrix $ \mathbb{E}\_{X \sim p\_{\text{data}}}[(X^T X )] $ , hence this part of the computation complexity is independent of the amount of calibration data! Due to the simple loss function, short back propagation chain, and the fact that the rotation matrix has only $d_{hidden}^2$ parameters, both the spatial and computational complexities remain small.
>
> Furthermore, the optimization of rotation matrices can be performed layer by layer sequentially, which further reduces the overall cost.
>
> The runtime for the training procedure, for LLaMA3-8B using 16 sequences of length 2048 as calibration data, requires only 8 minutes on an A100 when initialized with PCA without further training, and an additional 20 minutes for 100 training steps.

---

> ### Author Response · Authors · 2025-11-20
> **Response to Reviewer DQe5's Comments (3/4)**
>
> **Distillation**
>
> Your suggestion of using distillation to train rotation matrices is indeed insightful! Based on your guidance, we envision leveraging the densely activated model as the teacher and the sparsely activated model as the student, aiming for the sparse model to emulate the dense model.
>
> Our current approach actually embodies the idea of knowledge distillation, albeit indirectly, as it seeks to align the output of sparse activations with that of dense activations. As formulated in our main rebuttal:
>
> $$\min ||\text{Sparsify}(X)W - XW||^2$$
>
> Our proposed idea of "reducing inter-dimensional correlation" indirectly reduce this deviation, narrowing the gap between sparsely activated matrix (student) and densely activated matrix (teacher). Specifically, it:
>
> 1. Improve the accuracy of dimension importance estimation to prevent the erroneous removal of important dimensions.
> 2. Reduce the dimensionality of the activations, distilling information originally encoded in higher-dimensional vectors into lower-dimensional vectors, thus reducing the deviation caused by removed dimensions.
>
> Directly optimizing the rotation matrix or the entire model using the deviation as a loss function presents challenges. The difficulty lies in the sparsification function being discontinuous and non-differentiable, and determining the threshold ratio $\tau$ would require interleaving training, making end-to-end gradient-based optimization challenging and computationally more demanding than the current proposed method.
>
> Despite the implementation challenges, end-to-end optimization holds a higher theoretical potential and is certainly a worthwhile avenue for future research. We would be eager to explore this opportunity. Thank you for your suggestion!
>
> **Weakness 2: Ablation comparing "NorSA without rotations" vs "NorSA with rotations"**
>
> Thank you for this valuable suggestion which helps us improve the paper!
>
> In the **main rebuttal**, under section "1. Motivation of Using Rotation and Ablation Study", we show the large-scale ablation comparing "NorSA without rotations" vs "NorSA with rotations." In the tables, "NorSA w/o R" represents NorSA without rotations.
>
> Observations:
>
> 1. **NorSA w/o R outperforms TEAL:** The proposed norm-normalized threshold proves to be more effective.
>
> 2. **NorSA outperforms NorSA w/o R:** The advantage grows with increasing sparsity levels, indicating that reducing inter-dimensional correlations can enhance performance.

---

> ### Author Response · Authors · 2025-11-20
> **Response to Reviewer DQe5's Comments (4/4)**
>
> **Weakness 1: Conceptual Novelty**
>
> The conceptual novelty of this paper is highlighted in several aspects:
>
> 1. **Challenging the i.i.d. Assumption:** We question the i.i.d. assumption and propose considering the correlations between tokens and dimensions when applying activation sparsification techniques.
>
> 2. **Requirements for Activation Sparsification Functions:** In Section 4.1, Motivation, we present the requirements that activation sparsification functions should meet: Contextual Awareness, Computational Efficiency, and Flexible Parameter Allocation. We also provide reasons for why these requirements are essential.
>
> 3. **Reducing Inter-dimensional Correlations:** In Sections 4.3 and 5.2, as well as in the main rebuttal, we reveal through theoretical analysis and empirical validation that reducing inter-dimensional correlations enhances the performance of sparse activation models. There are many rotation matrices that can reduce inter-dimensional correlations, and PCA-based rotation is just one of them. We also propose a method to find these rotation matrices.
>
> 4. **Implementation and Verification:** We achieve the above objectives through a simple and efficient implementation (norm-normalized thresholds and rotations) and experimentally verify the effectiveness of these concepts.
>
> To the best of our knowledge, points 1, 2, and 3 are novel contributions in the field of sparse activation. They provide guidance and theoretical support for designing effective activation sparsification mechanisms. Existing works either rely on restrictive assumptions (e.g., TEAL) or do not offer clear and reasonable explanations for their methodology's effectiveness (e.g., La RoSA does not explain why PCA-based rotation is effective). Our paper contributes new and critical insights to the community.
>
> Point 4, which involves norm-normalized thresholds and rotations, serves as a simple yet efficient realization of the concepts in points 1, 2, and 3. It is important to note that this is not the entirety of the paper. Additionally, as you have praised in the strengths section, this implementation performs well and can be easily applied to existing systems. Achieving both usability and effectiveness is not trivial, and it is a notable highlight of our paper.
>
> For your questions, we have carefully prepared thorough responses. If you have any further inquiries, we are more than happy to address them. We hope our answers have effectively resolved your concerns, and we invite you to consider raising your score based on our clarifications.
>
> Thank you for your thoughtful review!

---

### Author Response · Authors · 2025-11-17
**Main Rebuttal**

Hello reviewers, we would like to express our heartfelt gratitude to all of you for your time and effort in reviewing our paper!

The acknowledgment of the empirical results achieved by our method and the recognition of the challenge to the independent and identically distributed (i.i.d.) assumptions in past work are greatly appreciated.

After carefully reading through each and every comment and suggestion, we address some common concerns below:

---

> ### Author Response · Authors · 2025-11-17
> **1. Motivation of Using Rotation and Ablation Study**
>
> One common point of feedback is the role of rotation presented in Section 4.3 and the large-scale ablation experiments.
>
> We briefly described the motivation and experimental effects of using rotation matrices in Sections 4.3 and 5.2. Below, we present the detailed impact of rotation matrices from both theoretical and empirical perspectives:
>
> **Theoretical Analysis**
>
> Improving sparse activation performance can, to some extent, be formulated as minimizing the difference between the outputs of the sparsified and original inputs:
>
> $$\min ||\text{Sparsify}(X)W - XW||^2$$
>
> This expands into the quadratic form:
>
> $$||(\text{Sparsify}(X)-X)W||^2 = (\text{Sparsify}(X)-X) WW^T (\text{Sparsify}(X)-X)^T.$$
>
> Let the index set of removed dimensions be $S$. The output difference caused by sparsification is:
>
> $$||(\text{Sparsify}(X)-X)W||^2 = \sum_{i \in S} X_i^2 \cdot WW^T_{i,i} + 2\sum_{i \neq j, i \in S, j \in S} X_iX_j \cdot WW^T_{i,j}.$$
>
> The difference in output caused by a specific dimension (e.g., i-th) removal consists of two terms: $X_i^2 \cdot WW^T_{i,i}$, and $\sum_{i \neq j} X_iX_j \cdot WW^T_{i,j}$ (which we refer to as correlation).
>
> Intuitively, a sparsification function that accurately estimates the impact of dimension removal (noted as dimension importance score) and thus removes the dimensions with lower impact will have better performance.
>
> The score formula:
>
> $$X_i^2 \cdot WW^T_{i,i} + 2\sum_{j \neq i} X_iX_j \cdot WW^T_{i,j}$$
>
> is evidently better than $|X_i|$ as an importance score. However, calculating it requires an $O(d_{\text{hidden}}^2)$ computational complexity (mainly because the correlation term), which is too expensive.
>
> Our idea is to approximate the accurate importance scores using a formula with $O(d_{\text{hidden}})$ computational complexity. This involves reducing the difference between $|X_i|$ (the importance score formula used in Section 4.2, which, due to the monotonicity of squares, is equivalent to $X_i^2$) and:
>
> $$X_i^2 \cdot WW^T_{i,i} + 2\sum_{j \neq i} X_iX_j \cdot WW^T_{i,j},$$
>
> specifically targeting the reduction of $\sum_{j \neq i} X_iX_j \cdot WW^T_{i,j}$. For implementation feasibility, we further simplified it to reducing $X_iX_j$.
>
> Therefore, Section 4.3's idea of reducing inter-dimensional correlation enhances and complements the approach detailed in Section 4.2 by improving the accuracy of dimensional importance estimation, and we have chosen rotation matrices as the method for implementation.
>
> Furthermore, beyond improving the sparsification function proposed in Section 4.2 for accurate dimension importance estimation, using rotation matrices to reduce inter-dimensional correlations is akin to reducing the dimensionality of activations/hidden states, essentially representing the same information with fewer dimensions. This aligns with the goal of sparse activation, which is to utilize only a subset of dimensions. For example, if a 10-dimensional vector can be losslessly represented by a 5-dimensional vector, this naturally achieves lossless 50% sparse activation.
>
> Therefore, rotation matrices that reduce inter-dimensional correlations can enhance the performance of sparse activation for two reasons: (1) they improve the accuracy of dimension importance estimation, and (2) they reduce the dimensionality of the activations.

---

> > ### Author Response · Authors · 2025-11-17
> > **1. Motivation of Using Rotation and Ablation Study**
> >
> > **Empirical Justification / Ablation Study of Rotation**
> >
> > To verify the general effectiveness of rotation that reduce inter-dimensional correlations, we conducted additional ablation experiments following the experimental setup of the main results section, across various models and sparsity levels. In the following tables, "NorSA w/o R" denotes NorSA without Rotation proposed in Section 4.3:
> >
> > **Language Modeling Performance**
> >
> > Perplexity measured on the test set of wikitext-2
> >
> > | Sparsity | Method | LLaMA3 8B | LLaMA3 70B | LLaMA2 7B | LLaMA2 70B | Mistral 7B | Qwen2.5 7B | Qwen2.5 72B |
> > |----------|-------------|-----------|------------|-----------|------------|------------|------------|-------------|
> > | 0%| Dense| 6.13 | 2.85| 5.47 | 3.32| 5.31| 6.85| 3.87 |
> > |||||||||
> > | 25% | TEAL | 6.37 | 3.95| 6.31 | 3.43| 5.53| 6.93| 3.93 |
> > | 25% | NorSA w/o R | 6.25 | 2.96| 5.57 | 3.39| 5.43| 6.92| 3.93 |
> > | 25% | La RoSA| 6.23 | 2.94| 5.51 | 3.34| 5.34| 6.90| 3.92 |
> > | 25% | NorSA| 6.19 | 2.91| 5.54 | 3.36| 5.33| 6.87| 3.89 |
> > |||||||||
> > | 40% | TEAL | 6.83 | 4.45| 6.40 | 3.61| 5.98| 7.20| 4.07 |
> > | 40% | NorSA w/o R | 6.65 | 3.49| 5.87 | 3.50| 5.94| 7.15| 4.10 |
> > | 40% | La RoSA| 6.60 | 3.37| 5.64 | 3.44| 5.44| 7.10| 4.06 |
> > | 40% | NorSA| 6.33 | 3.04| 5.59 | 3.40| 5.38| 6.97| 3.96 |
> > |||||||||
> > | 50% | TEAL | 7.56 | 5.61| 6.80 | 3.91| 7.17| 7.81| 4.31 |
> > | 50% | NorSA w/o R | 7.32 | 4.32| 6.41 | 3.72| 6.23| 7.56| 4.28 |
> > | 50% | La RoSA| 7.22 | 4.10| 5.87 | 3.62| 5.62| 7.42| 4.26 |
> > | 50% | NorSA| 6.57 | 3.33| 5.72 | 3.47| 5.46| 7.13| 4.07 |
> > |||||||||
> > | 60% | TEAL | 9.19 | 9.97| 7.82 | 4.53| 8.05| 9.99| 7.45 |
> > | 60% | NorSA w/o R | 9.02 | 6.03| 7.86 | 4.21| 7.84| 8.58| 4.95 |
> > | 60% | La RoSA| 8.57 | 5.51| 6.40 | 3.98| 6.04| 8.42| 4.90 |
> > | 60% | NorSA| 7.16 | 4.77| 6.02 | 3.63| 5.68| 7.49| 4.28 |
> >
> >
> >
> > **Zero-shot And Few-shot Performance**
> >
> > (Acc, MMLU)
> >
> > | Sparsity | Method | LLaMA-2 7B| LLaMA-2 70B | LLaMA-3 8B| LLaMA-3 70B | Qwen-2.5 7B | Qwen-2.5 72B| Mistral 7B|
> > |----------|-------------|------------------|------------------|------------------|------------------|------------------|------------------|------------------|
> > | 0%| Dense| 66.69, 45.85| 73.66, 68.80| 70.05, 65.26| 76.29, 78.71| 70.34, 74.21| 75.58, 86.08| 70.44, 62.34|
> > |||||||||
> > | 25% | TEAL | 65.94, 44.66| 73.31, 67.70| 69.40, 63.85| 73.23, 74.86| 69.76, 73.21| 75.05, 85.44| 70.06, 61.51|
> > | 25% | NorSA w/o R | 66.61, 45.61| 73.37, 68.39| 69.91, 64.62| 76.05, 78.59| 70.11, 73.78| 75.59, 85.75| 70.60, 61.80|
> > | 25% | La RoSA| 66.39, 45.66| 73.38, 68.74| 69.54, 64.85| 76.30, 78.13| 70.12, 73.74| 75.53, 85.62| 70.25, 61.81|
> > | 25% | NorSA| 66.52, 45.81| 73.71, 68.56| 69.91, 65.06| 76.28, 78.71| 69.97, 73.90| 75.57, 85.80| 70.49, 62.22|
> > |||||||||
> > | 40% | TEAL | 64.92, 43.46| 72.47, 66.78| 68.14, 59.84| 72.24, 73.23| 68.61, 71.44| 74.65, 84.80| 68.76, 60.17|
> > | 40% | NorSA w/o R | 65.39, 43.33| 73.34, 67.54| 68.78, 62.29| 73.92, 77.63| 69.22, 72.65| 75.44, 85.18| 69.72, 60.39|
> > | 40% | La RoSA| 66.15, 44.66| 73.31, 68.16| 68.79, 62.61| 75.41, 77.62| 69.67, 72.33| 75.35, 85.53| 69.44, 61.15|
> > | 40% | NorSA| 66.15, 45.42| 73.64, 68.22| 69.71, 64.07| 75.85, 78.29| 69.72, 73.35| 75.65, 85.83| 70.29, 61.23|
> > |||||||||
> > | 50% | TEAL | 63.22, 39.57| 71.92, 64.43| 64.92, 52.78| 70.80, 69.20| 67.76, 68.53| 73.74, 83.54| 66.73, 57.34|
> > | 50% | NorSA w/o R | 64.09, 40.33| 72.77, 66.28| 67.02, 58.95| 74.09, 76.61| 68.47, 70.29| 74.87, 84.95| 69.08, 58.19|
> > | 50% | La RoSA| 64.61, 43.10| 72.86, 67.57| 67.19, 58.65| 73.81, 76.51| 69.09, 70.09| 75.18, 84.34| 68.46, 58.80|
> > | 50% | NorSA| 65.49, 43.25| 73.31, 67.73| 68.82, 62.12| 75.34, 78.15| 69.26, 72.28| 75.45, 85.66| 69.84, 60.41|
> >
> > In almost all sparsity levels and models, NorSA (combining Sections 4.2 and 4.3) consistently outperforms NorSA w/o R, demonstrating the effectiveness of Section 4.3, with the performance margin notably increasing at higher sparsity levels.
> >
> > Thus, both empirical results and theoretical rationale substantiate the effectiveness of rotation matrices in our approach.

---

### Comment · Area_Chair_pCY8 · 2025-11-26
**Reviewer & Author Discussion**

Dear Reviewers,

We kindly encourage you to review and respond to the authors’ rebuttals. Your timely feedback is important for ensuring a fair and thorough review process. Thank you for your contributions to ICLR 2026.

Thank you very much for your time and support.

Best regards,

Area Chair pCY8

---

### Author Response · Authors · 2025-12-03
**Summary**

We sincerely thank the reviewers for their valuable feedback and the time and effort they have invested in reviewing our paper !

This paper presents a robust and easy-to-implement sparse activation mechanism. It leverages norm-normalized thresholds and correlation-reducing rotations to explicitly model correlations between tokens and dimensions, challenging the i.i.d. assumptions of previous approaches. The paper also provides theoretical analysis and an efficient, practical Triton kernel implementation for actual speedup.

We are encouraged to see that reviewers praised the strong performance of our method, surpassing previous approaches, the simplicity of the mechanism, which can be easily integrated into existing systems, and the reasonable challenge to the i.i.d. assumption.

A common concern among the reviewers was confusion about the role and principle of correlation-reducing rotations. We have addressed this in our main rebuttal with a detailed theoretical analysis and large-scale ablation study results.

For individual reviewers' concerns, please refer to our specific responses to each reviewer's comments for details.

We again express our gratitude for the reviewers' valuable suggestions.

Sincerely,

Authors

---

### Meta-Review · Area_Chair_AByM · 2026-01-06

**Summary:**

The paper proposes NorSA, a training-free sparse activation method for LLM decoding. It is based on zeroing out activations, and as a result reducing the amount of calculations and memory pressure on GPU. The method consists of 2 parts: i) normalization based thresholding and ii) rotation matrix to decorellated some dimensions. Reviewers agreed that the motivation is reasonable, the method is simple and practical, and the empirical results across multiple model families (LLaMA, Mistral, Qwen) consistently outperform prior methods such as TEAL and LaRoSA, especially at higher sparsity levels. However, several reviewers initially raised concerns about limited conceptual novelty, unclear motivation and role of the rotation component, and insufficient ablation separating the contributions of normalization versus rotation. Some concerns might have not addressed, and even if couple reviewers changed the score from 4 to 5, it will still be around the borderline without strong support from any reviewer. Unfortunately, I recommend this paper for rejection.

**Reviewer Concerns:**

Rebuttal provided multiple ablations requested by reviewers. Unfortunately, reviewers didnt participate in the discussion and didnt reply on new results. Authors did provide extra clarification on novelty related to SliceGPT and TEAL and improved experimental completeness. However, some concerns such moderate novelty and presentation quality might have not being answered.

**Reviewer Scores:**

The initial scores were 4,6,4,4 and after rebuttal, it is possible that some reviewers would change rating from 4->5.
DQe5 4 -> 5 because rotation motivation, ablations and novelty are clarified in the rebuttal.
L5Vk 6 -> 6 will keep the initial positive score.
mCoY 4 -> 4 will keep the score as it is based on novelty.
pMJ8 4 -> 4/5 as some ablations are added.

---

### Decision · Program_Chairs · 2026-01-26

Reject